



# Intraseasonal variability of wind waves in the western South Atlantic: the role of cyclones and the Pacific South-American pattern

Dalton K. Sasaki[1], Carolina B. Gramcianinov[2], Belmiro Castro[1,†], and Marcelo Dottori[1]

[1]Departamento de Oceanografia Física, Química e Geológica, Instituto Oceanográfico, Universidade de São Paulo, Praça do Oceanográfico, 191, Cidade Universitária, São Paulo, SP, Brazil

[2]Departamento de Ciências Atmosféricas, Instituto de Astronomia, Geofísica e Ciências Atmosféricas, Universidade de São Paulo, Rua do Matão, 1226, Cidade Universitária, São Paulo, SP, Brazil

[†]deceased, 11 October 2020

**Correspondence:** Dalton K. Sasaki (dalton.sasaki@usp.br)

**Abstract.** Extratropical cyclones are known to generate extreme significant wave height (swh) values in the western South Atlantic (wSA), which are highly influenced by intraseasonal scales. This work aims to investigate the importance of intraseasonal time scales (30–180 days) in the regional wave climate and its atmospheric forcing. The variability is explained by analyzing the storm track modulation due to westerlies winds. These winds present time-scales and spatial patterns compatible with the intraseasonal component of the Pacific South–American (PSA) patterns. The analysis are made using ECMWF's ERA5 from 1979 to 2019 and a database of extratropical cyclones based on the same reanalysis. Empirical orthogonal function (EOF) analysis of the 10m zonal wind and swh are used to assess the westerlies and waves regime in the wSA. The EOF1 of u10 presented a core centred at 45°W and 40°S, while the EOF2 is represented by two cores organized into a see-saw pattern with a center between 30°S–40°S and another to the south of 40°S. Composites of cyclone genesis and track densities, and swh fields were calculated based on the phases of both EOFs. In short, EOF phases presenting cores with a positive (negative) u10 anomaly provides a favorable (unfavorable) environment for cyclone genesis and track densities and, therefore, positive (negative) swh anomalies. The modulation of the cyclones track are significant for extreme values of the swh. The spatial patterns of the EOFs of u10 are physically and statistically consistent with 200 hPa and 850 hPa geopotential height signals from the Pacific, indicating the importance of the remote influence of the PSA patterns over the wSA.

## 1 Introduction

Surface wind waves (henceforth just called "waves") are relevant for a number of social-economic activities over the coast. They may impact the safety of operations in ports, oil shelves and also present relevant coastal impacts related to erosion and buildings damage (de Andrade et al., 2019). For instance, extratropical cyclones (hereafter, just "cyclones") are responsible





for drive most of the wave climate at mid and high latitudes, generating extreme conditions in the western subtropical South Atlantic (wSA) (Fig. 1), where significant wave heights (swh) may present values as high as 10 m (Gramcianinov et al., 2020c). The combination of waves and cyclones endangers the coastal population and society can benefit from its improved understanding and predictability.

      The wSA seasonal wave climate description depicts a region with wave parameters influenced by easterlies from the South

Atlantic Subtropical High (SASH) and frequent (3 to 5 times per month) S/SE wind associated with cold fronts (Pianca et al., 2010). In the wSA, cold fronts are synoptic features associated with cyclones, which can generate extreme waves due to its strong winds (e.g., da Rocha et al., 2003; Campos et al., 2018; Gramcianinov et al., 2020c). There are three main cyclogenetic regions in wSA coast: Southeastern Brazil (SE–BR, 25°S), Uruguay and Southern Brazil (LA PLATA, 35°S), and Central Argentina (ARG, 45°S–55°S) (Hoskins and Hodges, 2005; Reboita et al., 2010; Gramcianinov et al., 2019; Crespo et al., 2020).

The presence of high cyclone genesis activity impacts directly the wave climate in wSA, mainly from cyclones generated in the LA PLATA region (Campos et al., 2018; Gramcianinov et al., 2020c, 2021).

      Apart from the seasonal scales, a possible source of predictability of the wave climate could be related to the atmospheric interannual and intraseasonal variability. For instance, the North Atlantic Oscillation is a relevant interannual index that modulates the significant wave height (swh) in the North Atlantic (Dodet et al, 2010). In the South Atlantic, Pereira and Klumb-

Oliveira (2015) observed a significant but weak El Niño Southern Oscillation (ENSO) signal in the swh over the SE coast of Brazil. However, so far global studies of wind–wave interannual variability showed no significant relation between climate indexes such as the ENSO or the Southern Annular Mode (SAM) and the wave climate over the wSA (Godoi et al., 2020; Godoi and Torres Júnior, 2020; Reguero et al., 2015). On the other hand, there is a rise of evidence of modulation in intraseasonal time-scales of the wave fields by atmospheric processes (e.g Godoi et al. 2020a, Srinivas et al., 2020). As an example, the

Boreal Summer Intra Seasonal Oscillation (BSISO) in the tropical Indian Ocean is part of the summer monsoon and influences the surface winds, consequently modulating the swh anomalies over its different phases (Srinivas et al., 2020). Over the Pacific, Atlantic and Indian Oceans, Godoi et al. (2020) found significant dependences between the increase of swh anomalies and different ENSO–MJO phase combinations.

      In the South Atlantic, there are a few eligible atmospheric intraseasonal patterns that may influence the wave field. The

Pacific-South American patterns (PSA), for instance, are wave-trains in the geopotential height anomaly fields, which propagate from Eastern Australia to Argentina (Mo, 2000). PSA is originally known as interannual processes identified by two modes of the geopotential height: PSA1, with a dominant spectral peak between 36–40 months; and PSA2, with a dominant peak between 24–26 months. According to Mo and Paegle (2001), PSA1 would be associated with the low-frequency component of ENSO while PSA2 would be related to a Quasi–Biennial modulation of ENSO. More recently O'Kane et al. (2016, 2017)

have given a new perspective of PSA, showing evidence of the strong influence of synoptic-to-intraseasonal time scales. These authors performed a study to analyse the persistence of SH variability modes in a multiscale approach and found that PSA2 and another mode (PC5) are not well correlated with ENSO (O'Kane et al., 2017), as suggested before. O'Kane et al. (2016) found a quasi-stationary pattern in the South Pacific, which presents the same patterns as the PSA modes, but on synoptic-to-intraseasonal time scales. This signal has been associated with blockings on the Pacific and Atlantic and highlights the



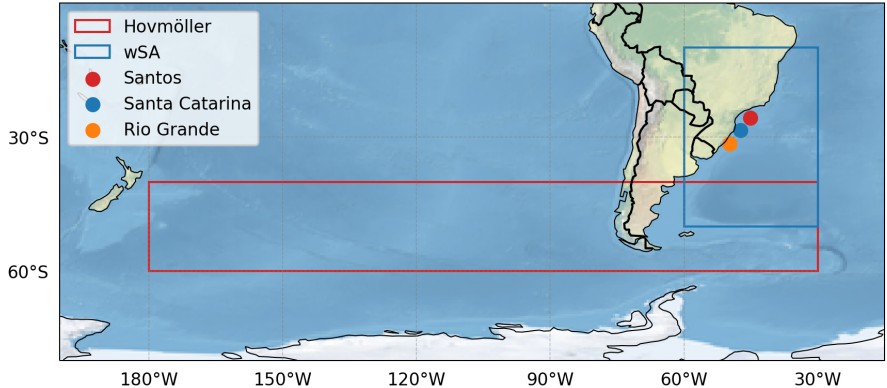

**Figure 1.** Domains and moorings used in this study: the wSA is the ocean area inside the blue rectangle; red rectangle shows the region used to calculate the Hovmöller diagrams (Sect. 3); circles show the approximate position of PNBOIA moorings used to evaluate the ERA5 swh results.

need for further investigation of PSA influences over South America, not only in terms of low-frequency but also considering intraseasonal time-scales.

The importance of intermediate time-scales in South America (SA) was also reported by recent works which identified the existence of an intraseasonal modulation of the atmospheric circulation, precipitation, and marine heatwaves (e.g. Paegle et al., 2000; Rodrigues et al., 2019). The SA dominant precipitation pattern in the summer is a dipole with centers of action over the

South Atlantic Convergence Zone (SACZ) and the subtropical plains (e.g. Paegle et al., 2000; Liebmann et al., 2004). Paegle et al. (2000) showed intraseasonal oscillatory modes with periods of 36–40 days and 22–28 days, both related to the Madden-Julian Oscillation (MJO). However, the last was also influenced by PSA-like wave-trains propagating from the midlatitude Pacific and turning equatorward as it crosses the Andes Mountains, which is supported by Liebmann et al. (2004). The observed change in precipitation is driven by circulation change: the low-level jet is enhanced or suppressed depending on the dipole

phase, which may also lead to a cyclonic disturbance development nearby 30°S–35°S region (Vera et al., 2002; Gramcianinov et al., 2019). In fact, patterns similar to PSA are strictly related to synoptic system propagation, being associated not only with precipitation, as well as with blocking, drought, and marine heatwaves (Rodrigues and Woollings, 2017; Rodrigues et al., 2019). The modulation of cyclones could have major implications for the ocean waves.

Therefore, the PSA can be considered a potential source of predictability in the wSA, influencing some meteo-oceanography

variables in the intraseasonal scale. For this reason, we hypothesise this variability mode also interferes in the wave field over the region. In this way, the questions that arise are: (1) Is the intraseasonal signal significant over the wSA regarding the wave field? If yes, (2) Which are the local forcing associated with this variability?, and; (3) Can this intraseasonal variability be linked to PSA? Since the surface wind field forces the wave field, they are used here as a proxy of the atmosphere variability, associated with the geopotential height at 200 hPa (Z200) and 800 hPa (Z850). In addition, once cyclones play an important

role in wave generation, cyclone tracks analysis is also applied.





The document continues with Sect. 2 describing the datasets and methods used in this work. Sect. 3 presents the results, showing the wind and waves variability in the intraseasonal time scale, the role of the modulation of the cyclones as regional forcing, and the analysis regarding PSA signal. The discussion is made in Sect. 4 followed by the conclusion in Sect. 5.

## 2  Data and methods

### 2.1  Datasets

The ERA5 reanalysis is a detailed reconstruction of the global atmosphere, land surface and ocean waves. It is the fifth generation of reanalysis produced by the European Centre for Medium-Range Weather Forecast (ECMWF; Hersbach et al., 2020) and made available through the Copernicus Climate Change Service (Copernicus Climate Change Service (C3S), 2017). This reanalysis is produced using 4D-Var data assimilation in ECMWF'S Integrated Forecast System (IFS), which is coupled with the wave model WAM (WAMDI Group, 1988). The atmospheric variables present 31 km ($0.28125°$) spatial resolution while wave parameters are in $0.36°$ resolution. Both atmospheric and wave data are available in hourly output from 1979 to the present (3-months delay). The data used in this study covers the period from 1979 to 2019, being the atmospheric variable in a $0.25°$ x $0.25°$ grid and the wave parameters in a $0.5°$ x $0.5°$ grid. An overview of the main characteristics of ERA5 can be found in Hersbach et al. (2019, 2020).

ERA5 presents improved fit for winds, temperature, humidity in the troposphere and also for the ocean wave height when compared to its predecessor, the ERA-Interim reanalysis (Hersbach et al., 2020). Belmonte Rivas and Stoffelen (2019) showed that ERA5 surface winds present an improvement of 20% relative to ERA-Interim, using ASCAT observations as verification, which is especially important since surface wind misrepresentation is one of the major sources of wind-wave modelling biases (e.g., Campos et al., 2018). Moreover, ERA5 is able to represent the Atlantic storm track distribution and characteristics, e.g., cyclone intensity (Gramcianinov et al., 2020a).

Regarding the wave field, Takbash and Young (2020) found that the reanalysis presents a good agreement in mean parameters and a global centennial wave height (swh[100]) magnitude and spatial distribution similar to the ones estimated by altimeter and buoy. Despite that, coastal regions are more susceptible to model systematic errors (biases) as result of the simplification of physical processes associated with shallow and intermediate waters interactions, and the inherited wind surface biases (e.g., Campos et al., 2018). Because of that an evaluation of the ERA5 wave parameters were made using buoy data. For that we used an array of moored buoys, which are part of the Global Ocean Observing System and Brazil's National Buoy Program (PNBOIA, see Pereira et al., 2017, for more details) and have measured meteo-oceanographic parameters since 2011. Among the entire data collection, three sites have better temporal availability: São Paulo, Santa Catarina and Rio Grande buoys. The evaluation showed a high skill of ERA5 swh between 2011 and 2018, both wave magnitude and variability. The results are presented in detail in the Sect. A (Fig. A1).



## 2.2 Cyclone identification and tracking

To analyse the influence of the synoptic systems in the wave climate variability we used the "Atlantic extratropical cyclone tracks in 41 years of ERA5 and CFSR/CFSv2 databases" (Gramcianinov et al., 2020b), which consists of a database containing cyclone track information from 1979 to 2019 for all Atlantic Ocean. The extratropical cyclones were tracked using the TRACK

algorithm (Hodges, 1994, 1995) following the method developed by Hoskins and Hodges (2002, 2005). The minimum duration and displacement of the cyclones are 24 hours and 1000 km, and only cyclones that spent a part of their life cycle within South Atlantic extratropical latitudes (85°S–25∘S, 75∘W–20∘E) are considered. More information regarding the database method and evaluation are available in Gramcianinov et al. (2020a).

Cyclone genesis and track spatial statistics, i.e., density, were computed using TRACK utilities applying the spherical kernel

estimator method (Hodges, 1996). Cyclogenesis density was built by using the position (latitude, longitude) where each cyclone was first detected by the algorithm, i.e., the first time-step of each track, while the track density considered the entire cyclone tracks. The significance for the differences of the cyclones spatial distributions between EOF/PC positive and negative phases (see Sect. 3.1) was calculated using Monte Carlo test (Hodges, 2008) with 1000 samples of the set of tracks for each phase.

## 2.3 EOF and wavelet analysis

Anomalies and filtered anomalies of swh, meridional 10-m wind (v10), and zonal 10-m wind (u10) were calculated from ERA5 dataset. The anomalies computations are obtained by subtracting the mean daily climatology from the original fields, while in the filtered results, anomalies were subjected to a Lanczos bandpass time filter with cutoff periods of 30 and 185 days. The selection of the cutoff period is based on the relevance of the intraseasonal variability indicated by the regional precipitation, marine heatwaves and PSA (O'Kane et al., 2017; Rodrigues et al., 2019) and on the results in section 3.1.

We use empirical orthogonal function (EOF) analysis to reduce the dimensionality of the data and decompose both anomaly and filtered fields over the wSA (Fig. 1). Removing the variability associated with periods smaller than 30 days in filtered anomaly fields avoided the inclusion of atmospheric meso and synoptic-scale propagating systems (e.g., cyclones and fronts). The first and second mode (EOF1, EOF2) produced by the analysis were selected and the results in section 3 show they are physically meaningful.

EOF analysis is usually applied over basin-size domains and captures features that correspond to large scale processes. In the Southern Hemisphere, for instance, the spatial signal of the Southern Annular Mode is present in the EOF of both seasonal swh (Hemer et al., 2010) and geopotential height fields (Mo, 2000). On the other hand, these large domains tend to separate the variability of large scale features in a given mode, which may attenuate (or even not show) the signature of a given regional process, such as the lobes in EOF maps in Figure 2 in https://doi.org/10.1175/MWR-D-16-0291.1 (O'Kane et al., 2017). For

this reason, we chose to restrict the EOF analysis to the wSA. Moreover, a wavelets analysis (Torrence and Compo, 1998) with a bias rectification (Liu et al., 2007) is used to identify relevant periods in the principal-component time series (PC) of the EOFs.



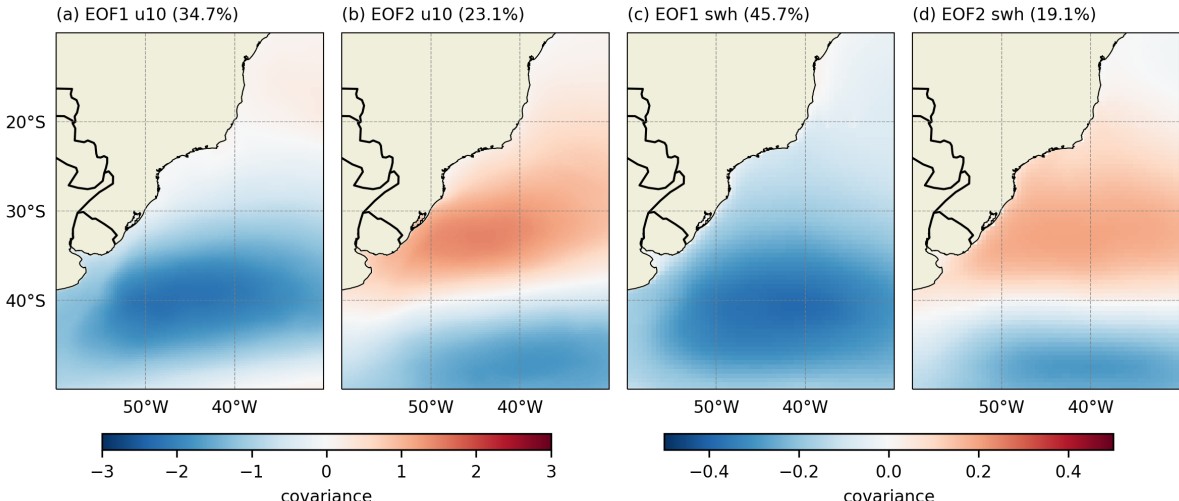

**Figure 2.** EOFs of filtered anomalies of (a,b) 10-m zonal wind (u10) and (c,d) significant wave height (swh) calculated on the wSA. The percentages represent the explained variance of each mode.

## 3  Results

### 3.1  Wave and wind spatio-temporal variability

Fig. 2 shows the first and second modes of the EOF of u10 and swh filtered anomaly fields. The EOF1 of u10 and swh (Fig. 2a,c) present similar spatial patterns with cores centred at 45°W and 40°S. The EOF1 of the u10 corresponds to 34.7% of the explained variance and the monopole is associated with strengthening/weakening of the wind anomalies. This variability may appear as an intensification, weakening or even reversal of the total u10 over the region. Similarly, the EOF1 of the swh (Fig. 2c) is also organized into a monopole, which retains 45.7% of the total explained variance. The EOF2 of the u10 and swh (Fig. 2b,d) consist of a dipole with centers between 30°S–40°S and to the south of 40°S and present an explained variance of 23.1% and 19.1%, respectively. The spatial patterns in the EOFs of the anomalies are omitted since they are very similar to the EOFs of filtered anomalies. The PC of the EOF1 of the anomaly and filtered anomalies of u10 and swh are presented in Fig. 3. These results exemplify the period from 2004 to 2008 and show the filtered fields retain a signal similar to the EOF of the anomaly fields. Results associated with the second mode of the EOFs of u10 and swh (not shown) present similar patterns to the first mode.

The values in the PC correlation matrix of the filtered u10, v10 and swh (Table 1) show correlation coefficients greater than 0.50 between the EOF1 of u10 and swh and also for EOF2, while the correlation between the EOF1 (EOF2) of u10 and EOF2 (EOF1) of swh present correlation values of 0.40 (0.21). Correlations values lower than 0.4 were not analyzed, which was the case of the correlation coefficients between EOFs of v10 and swh. The correlation values (highlighted in the table) indicate a





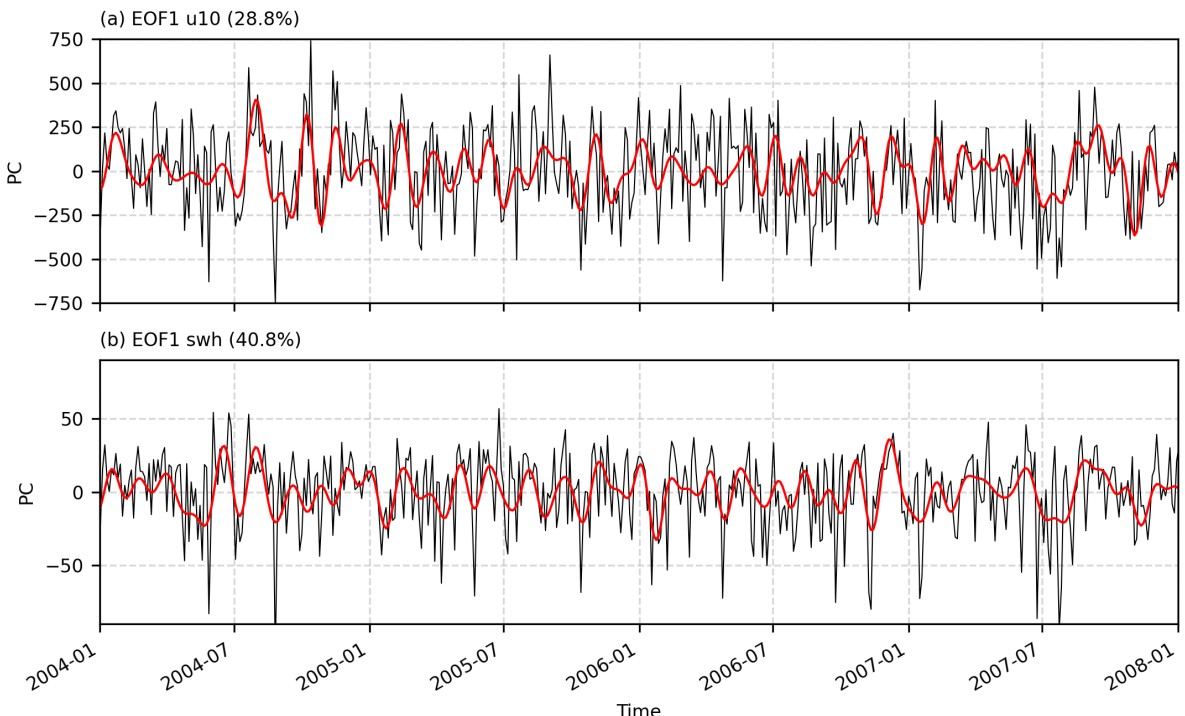

**Figure 3.** PC of the (a) EOF1 of u10 and (b) EOF1 of the swh. Black lines represent the PC of the anomaly fields and red lines represent the PC of filtered anomaly fields. The percentages represent the explained variance of EOF modes considering the anomalies fields.

**Table 1.** Correlation matrix between the EOFs of u10/v10 and swh filtered anomalies.

|          | EOF1 u10 | EOF 2u10 | EOF1 v10 | EOF2 v10 |
|----------|----------|----------|----------|----------|
| EOF1 swh | 0.50     | -0.21    | -0.25    | -0.31    |
| EOF2 swh | -0.40    | -0.52    | -0.37    | 0.12     |

direct intensification (weakening) of winds associated with greater (smaller) values of the swh, which are further evaluated in Sect. 3.3.

     Fig. 4 shows the wavelet power spectrum and global power spectrum of the PC considering the EOF1 of u10 and v10. Both cases present significant values ($p < 0.05$) only for the periods (T) between 1 and 6 months. Within this period band, the global wavelet spectrum of filtered and unfiltered PCs of the EOFs presents almost identical values. The analysis shows that in the
western subtropical South Atlantic, the intraseasonal variability of u10 and swh presents a significant signal, with the major peak at approximately 40 days. The results of the EOF2 in both u10 and v10 present similar results (not shown). Differences within 35 to 180 days period in the global wavelet spectrum plot may be justified by the different inputs in the EOF analysis, but the similarity of the PCs in Fig. 2 and the EOFs (not shown) indicate the filtered results are reliable.





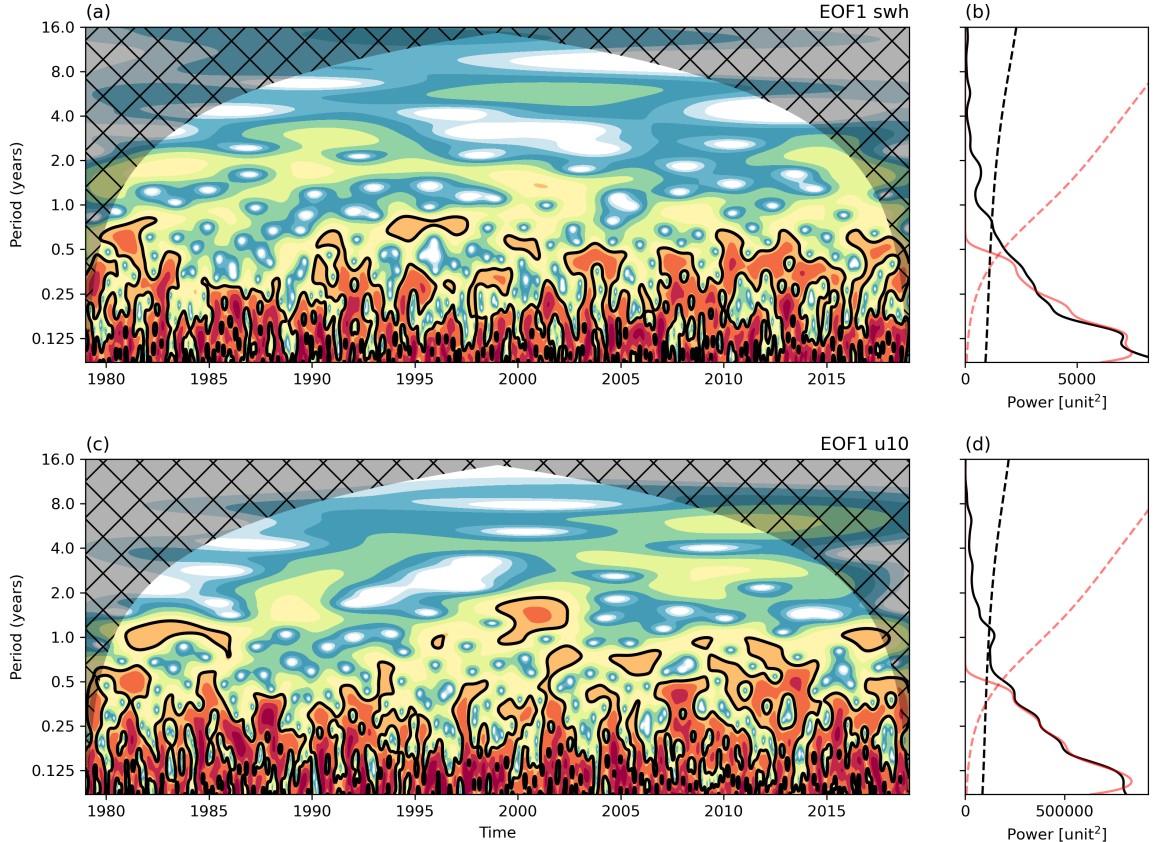

**Figure 4.** Wavelet power spectrum of the EOF1 of swh anomaly (a) and u10 anomaly (b), the black contours represent the 95% confidence level. Global wavelet spectrum of swh (c) and u10 (d) for the anomaly fields (black) and filtered anomaly fields (red). Dashed lines represent the 95% confidence level considering a red noise based on a univariate lag-1 autoregressive model.

In the following section, we built composites of the wave and wind fields based on the phases of EOF modes of u10 and swh,
in order to better understand the intraseasonal relationship between the variability of swh, cyclone genesis and track densities.
Fig. 5a exemplifies the definition of a phase in this study using the EOF1 of swh. For each EOF, Phase A (B) represents periods
when the PC values are greater (smaller) than 1 standard deviation. The spatial patterns of the EOF1 and EOF2 of both u10
and swh are a monopole and a see-saw pattern, respectively. The reconstruction (not shown) of the EOF1 during phase A (B)
leads to negative (positive) anomalies of the u10 and swh around $40°$S, while the reconstruction of EOF2 during phase A (B)
is associated with positive (negative) anomalies northward of $40°$S and negative (positive) anomalies to the south.



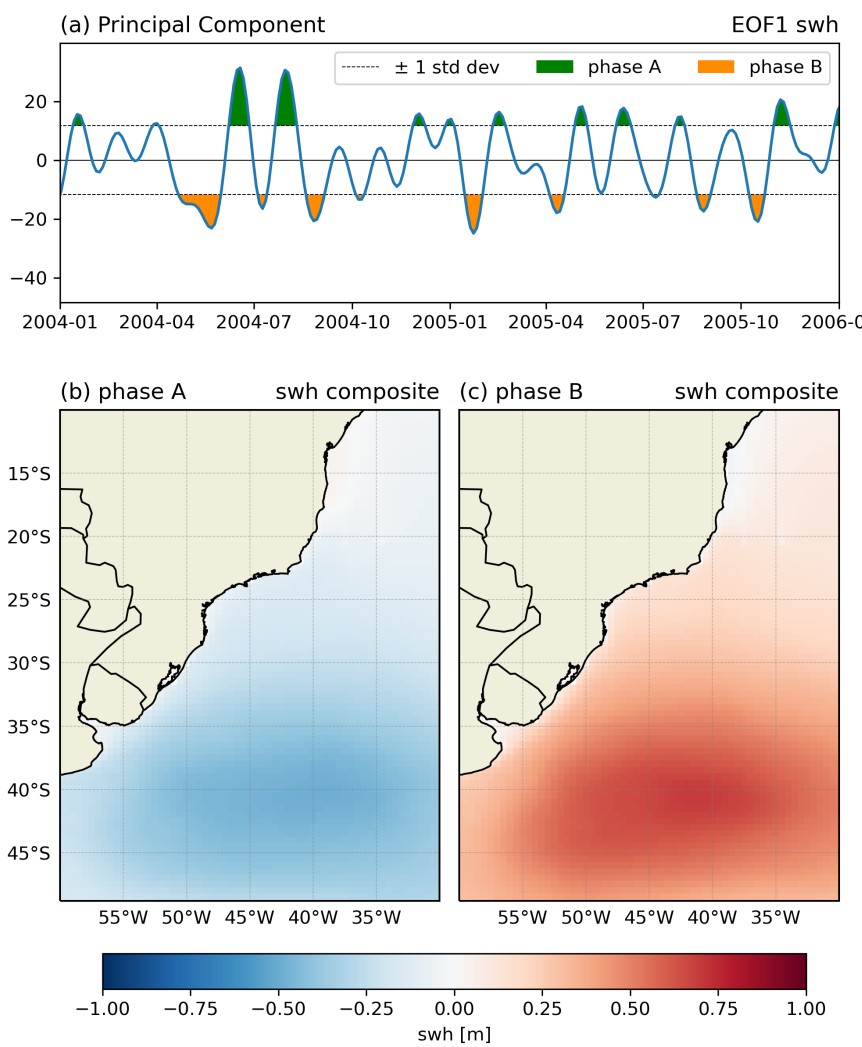

**Figure 5.** Time series of the EOF1 of (a) swh. Phase A (B) are periods when the time series presents values greater than (less than) +1 (-1) standard deviation. The composites are the average value of swh anomaly during (b) phase A and (c) phase B.

## 3.2 Cyclone genesis and track densities

The genesis and track densities differences between u10 EOFs phases (A minus B) are shown in Fig. 6, where regions with significant values are indicated by black dots. The track and genesis density for u10 EOF1 can be found on Appendix B (Fig. B1), where it is possible to evaluate the spatial patterns of density in order to understand the differences discussed in this section. The density differences computed for swh EOFs revealed similar patterns for u10, and, for clearance, they are presented on Appendix B (Fig. B2). The swh related fields are slightly weaker, showing a weaker response of swh EOFs phases, which is expected once this field is also influenced by remote forcing (i.e., swell). In Fig. 6, the spatial pattern of the





track density differences is represented by a zonal tripole in the EOF1 of u10, which distinguishes the regions over high, middle and subtropical latitudes, including the coastal area between Uruguay and Southern Brazil. The track density differences show

that phase B of EOF1/PC retains the main South Atlantic storm track between 40°S and 55°S, as reported in the literature (e.g., Hoskins and Hodges, 2005), resulting in fewer cyclones around Antarctica in a pattern similar to the one expected in the negative SAM phase (Reboita et al., 2015). In phase A, the opposite occurs, and the main storm track is shifted southward, resulting in fewer cyclones in middle latitudes. However, in these conditions, a secondary storm track emerges around 35°S. The cyclogenesis density differences reinforce this pattern, presenting in phase B less (more) genesis activity at Antarctica

Peninsula and more (less) activity between 40°S and 55°S, both in the ARG genesis region and the oceanic portion. The LA PLATA genesis region is enhanced in phase A, supporting the positive track density response at subtropical latitudes.

The track densities differences for EOF2 of u10 also reveal a zonal tripole, with narrower zonal bands when compared to EOF1 tracks (Fig. 6b). In phase A (B), the cyclone activity presents a positive (negative) signal between 30°S–40°S and 60°S-70°S, and negative (positive) between 40°S–60°S. By the genesis density, it is possible to see that genesis in LA PLATA and

northern ARG region are enhanced in phase A, as well in the Antarctica Peninsula. The genesis signal is mainly at the oceanic area adjacent to the South Brazilian and Uruguayan coasts, and separate the traditional ARG genesis region into two, being the northern portion active during phase A and the southern portion active in phase B. The behaviour observed in genesis and track densities can be interpreted as a direct consequence of EOF2 pattern once it shows a enhancing (weakening) of westerlies at 35°S (45°S) in phase A, affecting baroclinic waves development for the same reason explained for EOF1.

## 3.3 Waves Composites

Significant changes in the wave fields are known to occur in higher percentiles , as in cases associated with extreme events (Young et al., 2011). For this reason the composite calculations are based on a percentile approach. Figs. 7 and 8 present the differences between the composites of the swh $75^{th}$ percentile on phases A and B of the EOFs of u10 to ensure the EOF results represent real physical wind-sea interaction. The significance of the differences was calculated within the 95% interval and

determined by a bootstrap hypothesis test for the median of differences with 1000 realizations.

The swh composite difference shows that phase A (phase B) in the EOF1 of u10 is linked to negative (positive) anomalies in the swh fields southward of 35°S (Fig. 7). The difference computed for the phases of the EOF2 of u10 exhibit a see-saw pattern (Fig. 8), with positive (negative) values reflecting on the intensity of the wave anomalies in phase A (B). It is important to note that the composite differences are significant in a narrow area close to the central Brazilian coast (15°S-20°S), which

presents positive wave anomalies during phase A of u10 (EOF1). Also, the differences related to the EOF2 of u10 shows the $75^{th}$ percentile of the swh is relevant from northern Argentina towards the southern Brazilian coast. The signal in extreme wave composites follows the wind and wave behavior in each phase of EOFs discussed in the last section, given the similarity observed in Fig. 2, Fig. 7 and Fig. 8.

The relevance of the EOFs in the swh fields is further illustrated in Fig. 9, where composites of swh fields are determined

from high wave events (HWE) and the percentage corresponding to the significant height of wind-waves (shww) are presented. We define HWE as events where swh values at a point in the South Brazilian Bright (SBB; magenta point) exceed a 3 m



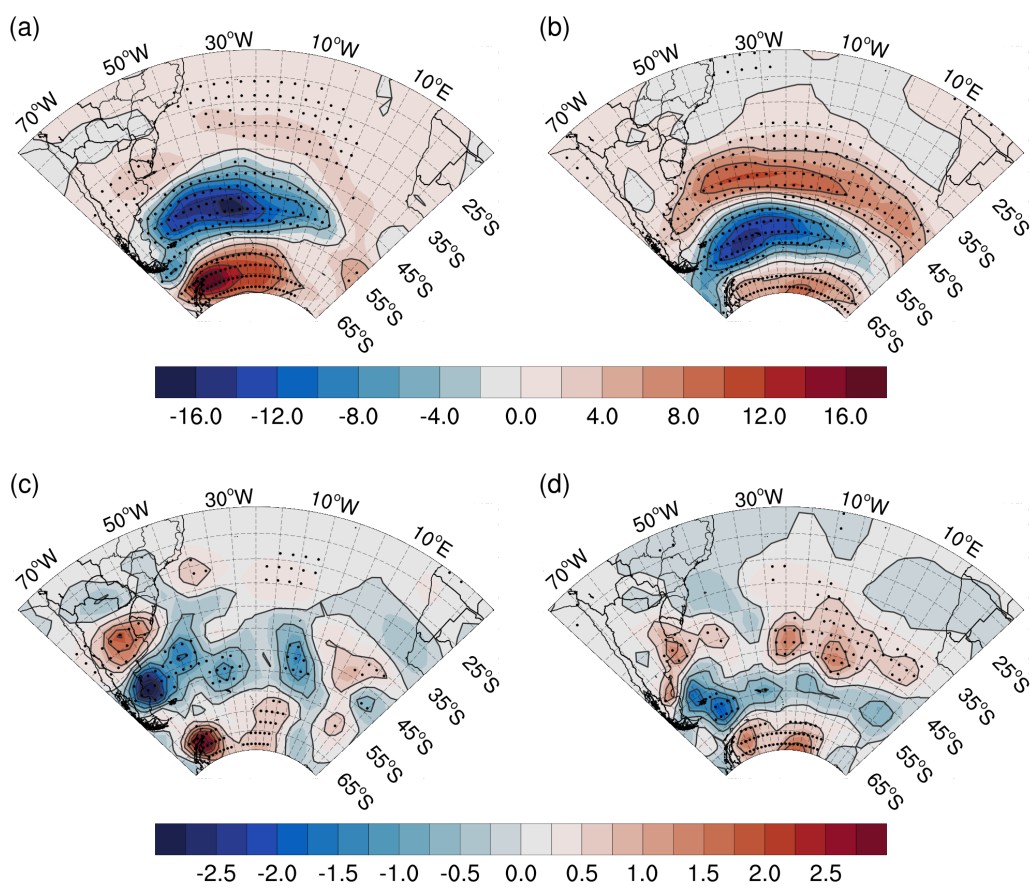

**Figure 6.** Cyclones (a,b) track and (c,d) genesis density differences between phases A and B of (a,c) EOF1 and (b,d) EOF2 of u10. Differences are calculated using phases A minus B and are also contoured by 5 density unit intervals for the track and 0.5 for the genesis. The black dots indicate where the difference is significant within a 95% confidence level. Density unit is the system per year per area, where area is in $10^6$ $km^2$ ( $5°$ spherical cap).

threshold while occurring concomitantly with at least one active cyclone at the selected domain (i.e., wSA). These composites are also built according to the different phases of u10 EOF modes.

A similar number of HWE (n∼80) was identified during phases A and B of the EOF1 of u10 (Fig. 9a,b). The maximum
mean swh (>3.5 m) in phase A is centred at approximately $30°S$ and $45°W$, in the SBB region, being 60% of these events associated with the shww. In phase B, high mean swh values are more distributed within the domain, following an NW-SE orientation, from the SBB region to the SE domain's boundary. Over the domain, the shww corresponds to 40–50% of the swh. Both phases present a similar spatial distribution of cyclone centers, which are most often positioned to the southeast of the SBB, but cyclones are scattered also to the south and southwest of the SBB region. The effect of the wind composites show




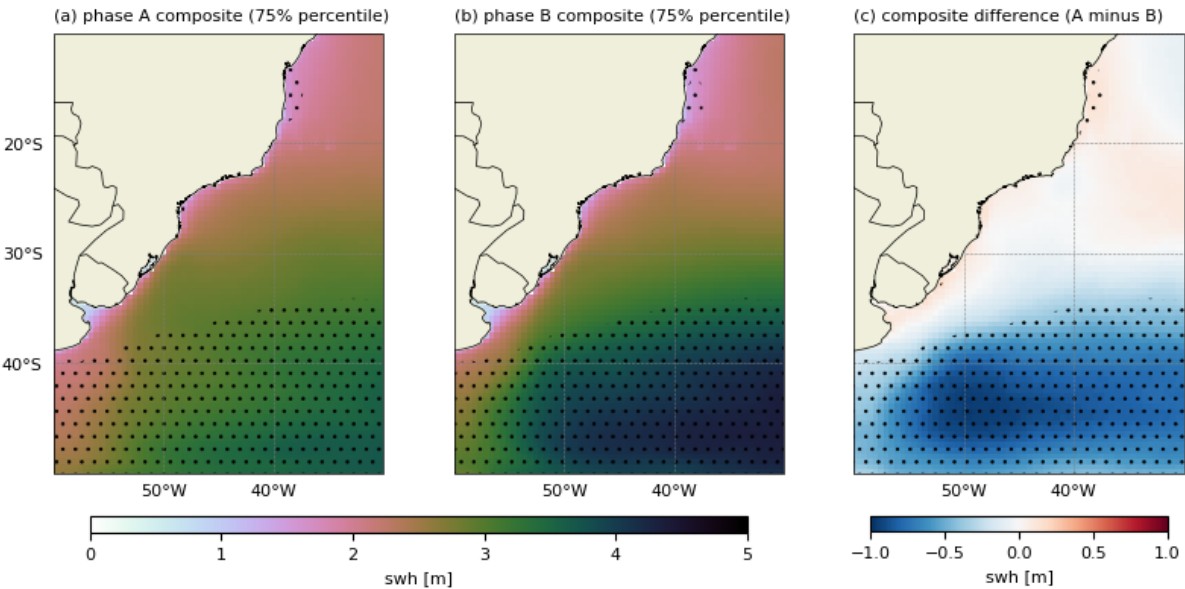

**Figure 7.** Composites (75th percentile) of swh calculated using phases (a) A and (b) B of the EOF1 of u10 and (c) difference between the composites. The black dots represent the areas where the difference between phases are within a 95% confidence level.

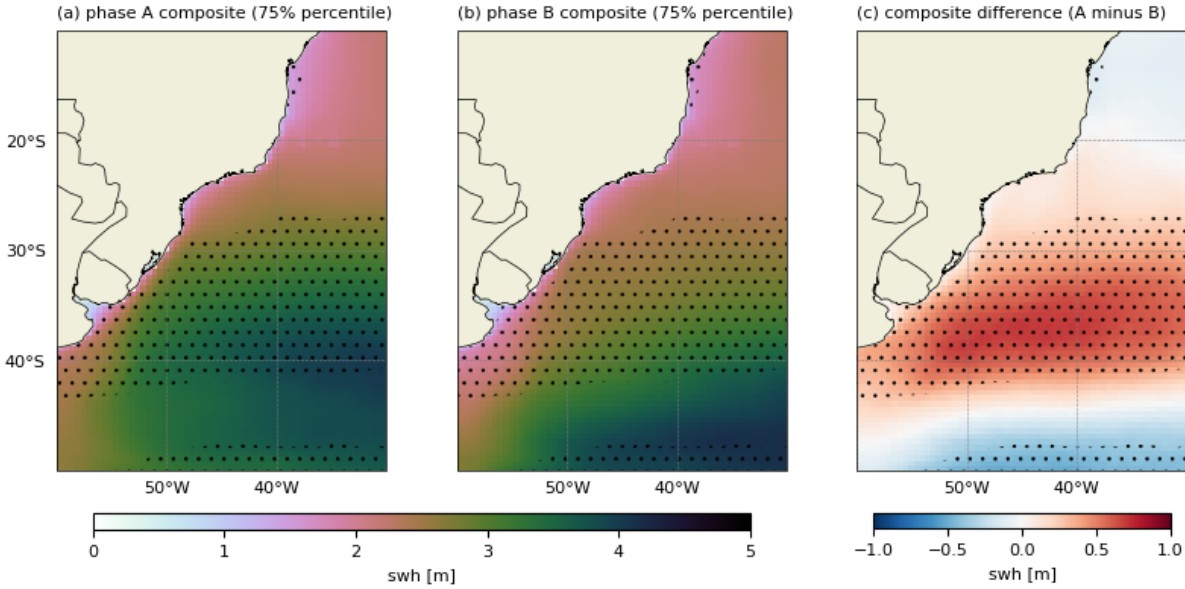

**Figure 8.** Composites ($75^{th}$ percentile) of swh calculated using phases (a) A and (b) B of the EOF2 of u10 and (c) difference between the composites. The black dots represent areas where the difference between phases are within a 95% confidence level



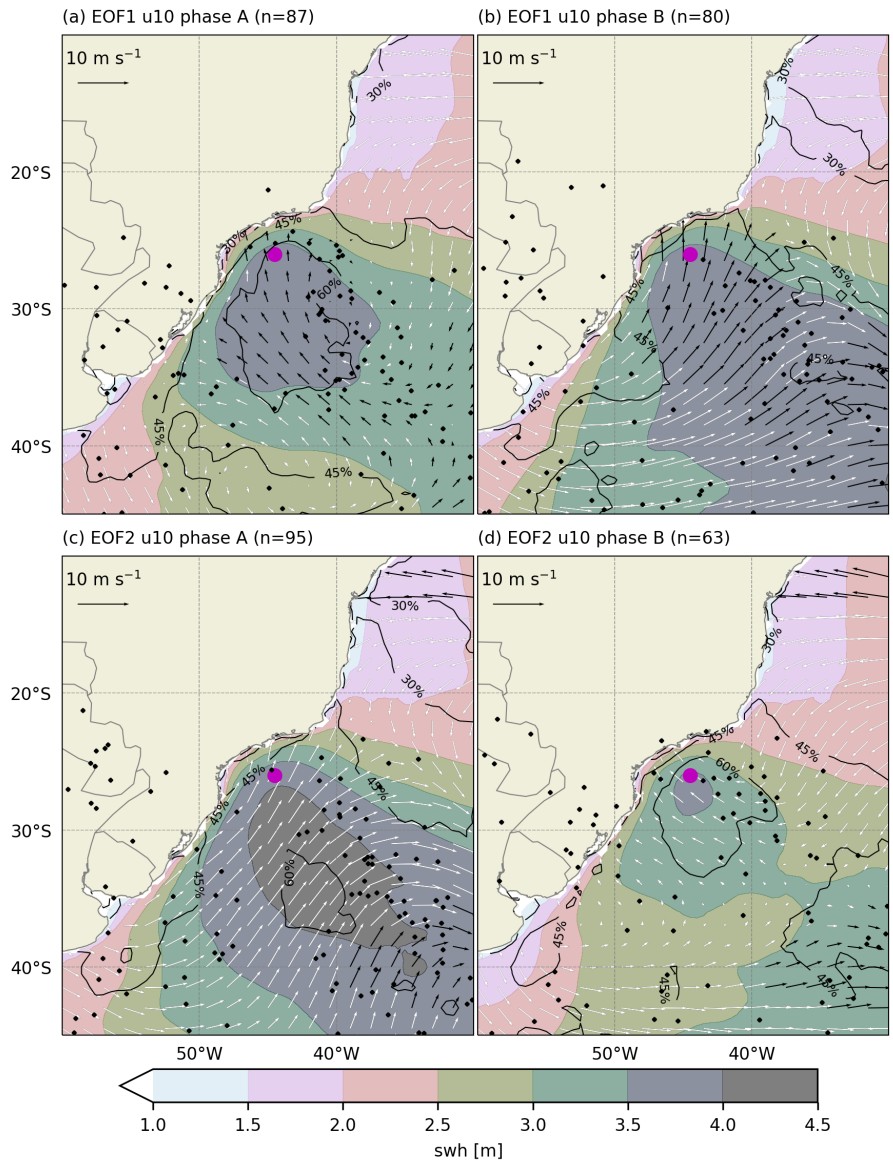

**Figure 9.** Composite of the total swh (shaded) for the high wave occurrence in the SBB (purple dot; 30°S, 45°W) in the phases (a) A and (b) B. The sea-wave height is shown as percentage from the swh in contour lines and the 10m-wind is indicated by the vector fields. White vectors are areas where the differences of swh between phases of a given eof are within the 95% confidence level.

the wind fields resembling a cyclone centered at 30°S and 38°W during phase A, whereas a trough-like pattern indicates a cyclonic circulation (smoothed by the composite) eastward from SBB in phase B.

In the composites of phases A and B of the EOF2 of u10 (Fig. 9c,d), 95 and 63 HWEs were identified, respectively. Phase A is associated with a larger HWE number and also presents cyclones that occur mainly to the southeast, but also to the





southwest of the SBB. In this phase, swh values range from 3.5 to 4.5 m in a band oriented NW-SE from the SBB, where the

shww corresponds to 60% of the swh. The lower number of HWE in phase B agrees with general lower swh values presented

in the composite, with shww percentage corresponding to 40–50% of the total swh. In this phase, the exception occurs over

the SBB region, where the swh values range from 3.0 to 4.0 m and the percentage of shww is approximately 60% of the swh.

Here, the trough in the wind vectors is present in phase A, while a regional cyclonic wind pattern is centered near 25°S and

42°W in phase B.

## 3.4   Evaluation of remote signal

Up to this point, we have shown that the EOFs of u10 within the intraseasonal band are significant for the variability of cyclones

and waves in the wSA, especially regarding the extreme wave climate. The remaining question is: Is this variability pattern a

part of a larger organized system? Over the wSA, the spatial patterns revealed by the EOFs of u10 are similar to features of the

PSA found using EOFs of the geopotential height (Irving and Simmonds, 2016). There is little agreement to what EOFs of the

geopotential height represent to the sub and extratropical environment between the South Pacific and South Atlantic — some

associate it to SAM and PSA (Ding et al., 2012), others relate it to an eastward propagating wave train that may be connected to

MJO influence on the western South Pacific (Paegle et al., 2000; Liebmann et al., 2004; Irving and Simmonds, 2016). However,

we will not consider here the effect of MJO, since its effects are observed only during austral summer (e.g., Liebmann et al.,

2004; Rodrigues and Woollings, 2017). Following the discussion and results presented by O'Kane et al. (2017), except on

the summer, the subtropical jet works as a barrier to waves propagating from the tropics (e.g., Hoskins and Ambrizzi, 1993;

Ambrizzi and Hoskins, 1997), reducing the influence of MJO in the SH extratropics. Restricting the analysis to a unique season

would reduce the sample size and would eliminate seasons where the cyclones and wave climate are more severe in the region

(e.g., Pianca et al., 2010; Gramcianinov et al., 2020c).

PSA modes evaluated by EOFs in the South Pacific and South Atlantic domains require multiple EOF modes (geopotential

height, for instance) to depict the wave train signal in the atmosphere that extends from the central South Pacific to the South

Atlantic (O'Kane et al., 2017; Irving and Simmonds, 2016). In contrast with the usual approach of using the geopotential height

fields, the evaluation of the signal that propagates from the South Pacific into the wSA is made using Hovmöllers diagrams.

These diagrams do not separate the variability in different modes, but provide valuable insights into the associated variability

that are difficult to interpret using the EOF approach.

The PSA patterns occur across the South Pacific and South Atlantic domains, but the analysis of EOF in such a large domain

would interfere and smooth the variability signal observed in the wSA. To evaluate the spatial distribution of the PSA patterns,

usually observed in the Z200 and Z850 fields, we used composites during the different phases of the EOF of u10.

Fig. 10 illustrates a wave-like spatial signal in both geopotential height fields, which extends from 150°W in the South

Pacific to 20°W in the western South Atlantic. The composites of Z200 and Z850 based on phase A of EOF1 of u10 present

multiple cores between 40°S and 60°S, alternating positive and negative values with an opposed signal to phase B. The stronger

signal in Z200 and Z850 occurs in the South Atlantic in both phases and overlaps with the monopole observed in the EOF1 of



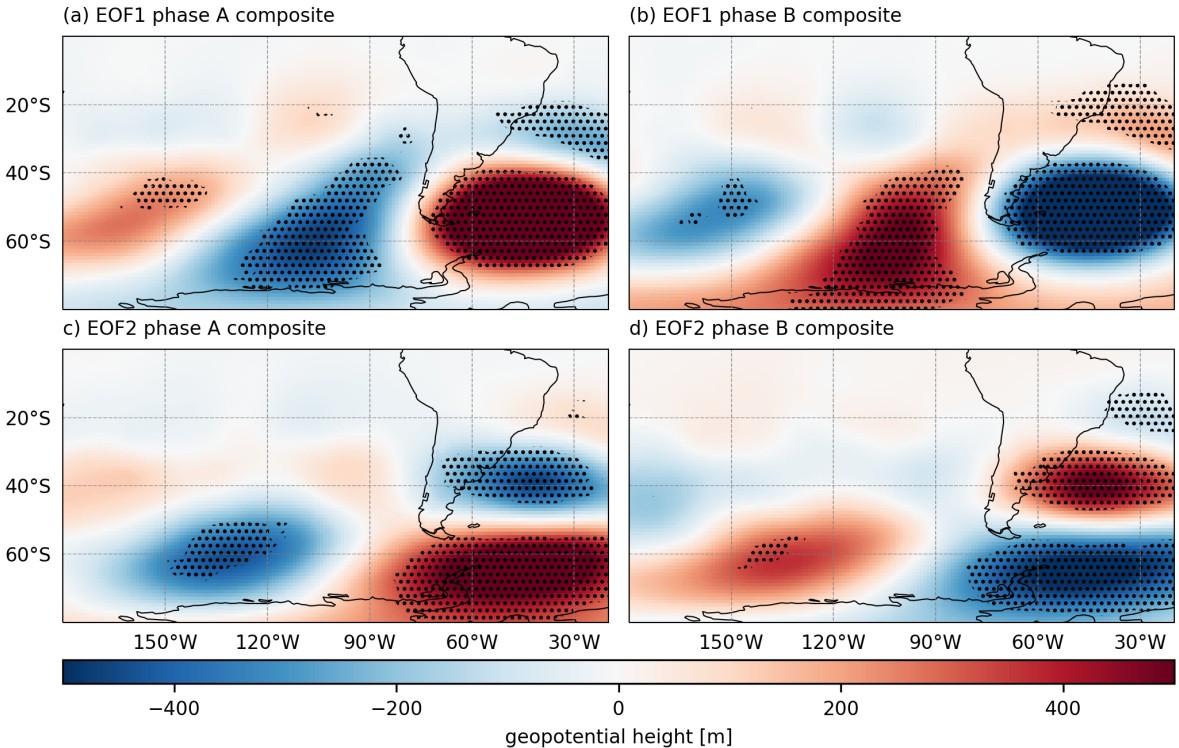

**Figure 10.** Composite of the geopotential height anomaly at 200hPa (shaded) for the u10 EOF1 in the phase (a) A and (b) B, and EOF2 in the phase (c) A and (d) B. Black dots denote where the signal is significant within the 95% confidence level.

u10 (Fig. 2a) southward of 40°S. The second most intense core extends from the Amundsen Sea in Antarctica (120°W–90°W) towards the western coast of South America at 40°S.

In both phases of the EOF2 of u10 an alternating pattern is also present in the Z200 and Z850 composite fields. The phase A (B) presents a core with local maximum (minimum) in geopotential height fields on the South Pacific northward of the region (150°W-120°W) between the Ross Sea and Amundsen Sea. Over the South Atlantic, there are two main cores with a meridional distribution, with opposite signals between the phases. The first is located southward of 60°S and the second is centered at 40°S, between the dipole pattern presented in the EOF2 of u10 (Fig. 2b).

Considering that single EOF modes analysis are inappropriate for the evaluation of propagating signals, we used Hovmöllers diagrams in the assessment of the PSA wave-train propagation in the Z850 and Z200 filtered fields (30–185 days). The Hovmöller diagrams, presented in Fig. 11, are calculated in the area (meridional average) indicated in Fig. 1 and show patterns of variability from the period n between 1988 to 1996. The time interval choice is arbitrary and aims only to exemplify the signal propagation between the Pacific and South Atlantic domains. The Hovmöller diagram of Z200 shows a complex pattern of intraseasonal variability, where part of the signal propagates eastward over time and shows wave-like features, such as the

presence of anomalous troughs and ridges system, i.e., sequential positive and negative values. In some cases, the signals appear



**Figure 11.** Hovmöllers of the filtered Z200 anomaly signal from 1988 to 1995 and retains the variability within 30–185 days with a Lanczos bandpass filter. The green line indicates the longitudes corresponding to the wSA. The subscripts 1 (2) in the months labels corresponds to the first (second) year indicated in the title of each panel.

at $180°$W and reach $30°$W over a period of 3 months (e.g., January–March 1988, July–September 1989, January–March 1994). In other cases, the signals start near $160°$W and last approximately 6 months (e.g., November 1988–March 1989, November–Marc, September 1992–January 1993). Other features that can be noted are the westward propagating signals (e.g., September 1991–January 1992, January–March 1993), revealing that not all intraseasonal variability in wSA is necessarily connected with the eastward propagating features. This is particularly noticeable (e.g., from September 1994–July 1995) when the propagating signals are absent from the diagram.





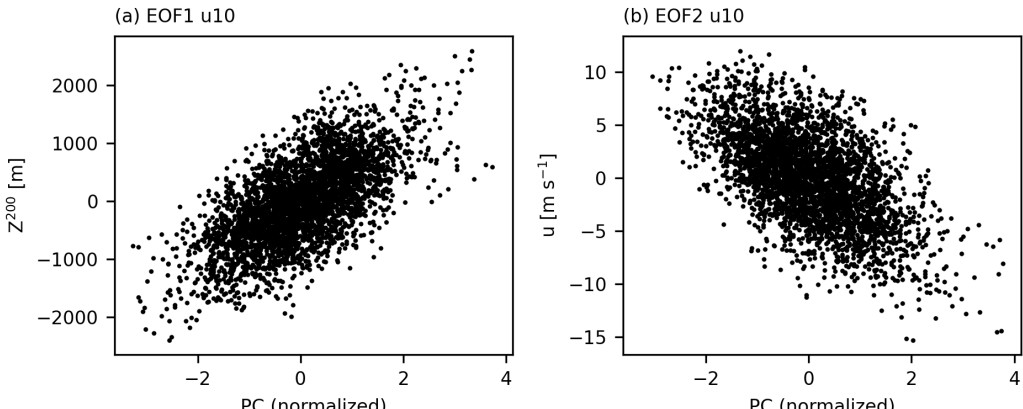

**Figure 12.** Scatter plot of the time series of the (a) EOF1 of u10 and Z200 and (b) EOF2 of u10 and u10, with correlation coefficient values corresponding to 0.67 and -0.59, respectively. The time series of Z200 and u10 were calculated by taking the spatial average at each time step in the intersection of the areas used to calculate the EOFs and Hovmöllers diagrams.

## 4 Discussion

### 4.1 Intraseasonal variability in wSA

The frequency spectrum showed the relevance of the intraseasonal frequency in the u10 over the wSA (Fig. 4). In the surface
wind field, the pattern represented by EOFs may correspond to the westerlies intensity and meridional variability forced by the mid-latitudes upper-level jet changes (e.g., Hare, 1960). The EOF1 of u10 represents the strengthening (weakening) of the westerly wind belt around 30°S and 45°S in phase A (B). In this way, phase A of EOF1 of u10 is associated with lower swh values, which also consists of periods with a less active storm track in the South Atlantic located between 40°S and 55°S (Fig. 6a). On the other hand, the weakening of the surface winds, observed in phase B, may play a role in the inhibition of baroclinic
wave growth, as long as it reflects directly to the up-level jet and baroclinicity (Hoskins and Ambrizzi, 1993; Ambrizzi and Hoskins, 1997),. In this situation, fewer and less intense cyclones are expected to develop in the region representing, thus, inefficient wave generation mechanisms. The EOF2 patterns, representing a meridional shift of the westerly wind and its effect on the cyclone patterns and wave fields, is analogous to the EOF1 explanation. Over different phases of the EOF2 of u10, there is an enhancement (weakening) of the cyclone activity between 30°S-40°S (Fig. 10b) in phase A (B) of the EOF2 of u10,
which leads to increase (decrease) of swh values in the northern zonal band of the see-saw pattern represented in (Fig. 2d). This is evidenced by both EOF2 of u10 and swh, which exhibit an approximate superposition of the dipole structures (Fig. 2b,d). In this way, positive (negative) anomalies of u10 linked with the EOF modes are associated with increases (decreases) in the swh anomaly fields. This relation is also reinforced by the positive correlation values computed for the PCs (Table 1) The same pattern is observed in the composites considering the 75th percentiles of the swh (Figs. 5,8). However, the percentile





approach allowed a further view of how the intraseasonal modes would interfere in extreme wave climate. The changes in swh go up to 1 m, which is a high value in an extreme analysis perspective. Specially regarding regions with significant changes in economically strategic locations of the South American coast, such as central Brazilian coast and northern Argentina, Uruguay, and the southern Brazilian coasts, for modes represented by EOF1 and EOF2, respectively.

**4.2   Impacts on extreme waves regional climate**

The composite analysis described above provides a general view of the processes, but in order to have a more in-depth idea of what is happening in a specific region and include the cyclones dataset into the analysis, we evaluate the composites of high-wave events (HWE) (Fig. 9). The selected region of this analysis lies purposely outside of areas with significant differences, which shows that the area of influence of the EOFs of u10 are not restricted to the significant areas represented in Figs. 7 and 8.

During EOF1 of u10, the similarity in the number of HWE is consistent with the fact the cyclone tracks associated with the EOF1 of u10 are mainly concentrated to the south of 40°S, which makes this area less likely to be affected by a given phase. In phase A we see high swh values concentrated over SBB region due to the increase in genesis in LA PLATA (Fig. 7), but for cyclones with small spatial range,i.e., short tracks (Fig. 6a). On the other hand, phase B presents an enhanced cyclonic activity between 40°S and 55°S, associated with the increase in genesis in the ARG region (Fig. 6d). This cyclone behavior justify the

spreaded swh pattern in phase B composites (Fig. 9b), once its contribution to the wave field in the region is also linked to its variable spatial scale (1500 - 2000 km radius) and swell propagation. In fact the shww percentages in the composites show that the remote effect in the total wave field is larger in phase B than in A.

The EOF2 of u10 presents a larger number of HWEs during phase A and this is a consequence of the higher genesis and track densities between 35°S–45°S. In phase B, more cyclones are generated southward of 45°S, which lead to lower and

less intense HWEs. However, a small region over the SE Brazilian coast (30°S) presents more genesis (negative values in Fig. 6d), indicating a wind forcing source close to SBB. The cyclones generated in SE–BR are usually weaker and short-time systems, which can generate HWE but not as much as the LA PLATA region (Gramcianinov et al., 2020c). Therefore, the swh composites are still presenting relatively high sea-wave percentages, although the swh are lower than the ones presented in the composite of phase A. One can note that the proximity of active cyclogenesis region influences directly the percentage of

sea-wave relative to the total swh. However, the percentage never crossed the 60% value, revealing a high influence of remote forcing in the SE Brazilian coast wave extremes.

Gramcianinov et al. (2020c) evaluated extratropical cyclones environments in the wSA and found three situations where extreme waves are generated: (1) west/southwestward of the cyclone center, behind the cold front; (2) north/northwestward of the cyclone center, ahead of the cold front, and; (3) eastward of the cyclone center, along the warm front. The most commonly

observed case in the composites is the situation 1, which is identifiable by the amount of cyclone centers placed to the east of the reference point (SBB) and the trough in the mean wind field. Situations 2 and 3 are more difficult to identify but are likely to be associated with the cyclone centers over the ocean occurring to the south and southwest of SBB, and also over land.





### 4.3 South Pacific forcing

The composites of the geopotential height fields for different phases of u10 EOFs show a wave-train coming from South Pacific
mid-latitudes to wSA, similar to PSA as described in the multi-scale approach of O'Kane et al. (2017). The intraseasonal signal
over wSA is consistent with the features presented in the Hovmöller diagrams (Fig. 11), where the PC of the EOF1 (2) of u10
presents an approximate linear response to the mean time series of the Z200 (u10) (Fig. 11). The area used to calculate the
time series of Z200 does not overlap completely with the wSA box, but it captures the monopole of the EOF1 of u10 and the
southern core of the dipole of the EOF2 of u10. Eastward propagating from the South Pacific signals organized into wave-train
features.

The key point of the Hovmöller diagrams is the fact these eastward propagating systems can appear at some occasions
in the South Pacific to the west of the 90o meridian one month before reaching the wSA. The MJO variability is known to
contribute to austral summer synoptic features over South America such as South Atlantic Convergence Zone and atmospheric
blockings (e.g., Liebmann et al., 2004; Rodrigues and Woollings, 2017) and could force part of the eastward propagating
signals in the Hovmöller diagrams with the origin to the west of 18°W. Moreover, other eastward propagating systems are
present and start in between 160°W and 90°W. These cases may arise as the PSA modes associated with local disturbances and
the internal waveguide dynamics over midlatitudes (O'Kane et al., 2017). As discussed by O'Kane et al. (2017), apart from
the summer, the subtropical jet reflects and breaks stationary Rossby waves from tropical sources, as shown by the ray tracing
theory (Ambrizzi et al., 1995; Li et al., 2015). In this way, the PSA pattern in the intraseasonal frequency consists of stationary
Rossby waves generated by the subtropical and polar jets internal instabilities, with little evidence for sustained equatorial
tropical sources. Since the connection between the observed PSA pattern and EOF modes of u10 exists, these results indicate
these wave propagating regimes impact cyclone genesis and track, as well as the associated wave fields, especially regarding
the extreme waves.

### 5 Conclusions

This work aimed to investigate the existence and effects of the intraseasonal variability in the wave field over the wSA. In order
to accomplish this goal, we established three main questions: (1) Is the intraseasonal signal significant over the wSA regarding
wave field? (2) Which are the local drivers associated with this variability? (3) Can this intraseasonal variability be linked to
PSA? Regarding the first question, we found evidence of significant variability of the wave field in the wSA in intraseasonal
time-scales, with a dominant period of 38 days. This variability occurs in response to changes in intensity and position of the
westerlies within the domain. The strength and shifts of the surface winds also lead to a modulation of cyclone tracks and
genesis, which is strictly related to wave generation in wSA. In fact, the analysis of high percentile composites in each mode
variability showed that the extreme wave climate is highly impacted by the intraseasonal scale, presenting variations up to 1 m
of swh. This finding is relevant once the wSA is an economically active area, with many oil platforms and active ship routes.
Such naval structure and activities demand high-quality forecasts and metocean analysis, which can benefit from the inclusion
of more reliable intraseasonal time-scale processes.



According to past studies that analysed other variables (e.g., precipitation, sea surface temperature), one of the sources of intraseasonal variability in South America is the PSA (O'Kane et al., 2017). The MJO influence is also documented (Liebmann et al., 2004; Rodrigues and Woollings, 2017), however its effects are usually observed in the summer. Guided by the findings of O'Kane et al. (2017) we considered that PSA in intraseasonal scales during other seasons are led by internal variability of the

subtropical and polar jets, excluding the influence of tropical convection, and consequently MJO. We analysed the wave-train pattern present in the geopotential at mid and upper levels and found a significant link between them and the intraseasonal variability modes presented for wave and wind field EOFs in the wSA domain. In our results, the signal of the wave train propagation from the South Pacific to the wSA occurs mainly eastward within 3 and 6 months. Also, we found a less frequent westward propagating signal, which, although being interesting, does not represent the PSA pattern.

Although we focused on the impact of the intraseasonal variability on the wave field, our analysis was restricted to wave height. In this way, it would be valuable to understand the impact of these intermediate time-scale on other wave parameters, such as peak period, mean wave direct and others. Further investigations about the conditions that allow the propagation of this intraseasonal signal from the Pacific to Atlantic seems to be valuable, given the large controversy regarding PSA time-scales and sources.

*Code availability.* The codes will be available to the reader, upon a reasonable request by email.

*Data availability.* The ERA5 products were generated using Copernicus Climate Change Service Information [2021] (https://cds.climate.copernicus.eu/, last access: 20 January 2021) (Copernicus Climate Change Service (C3S), 2017). The Cyclone tracks used in this study were obtained at "Atlantic extratropical cyclone tracks in 41 years of ERA5 and CFSR/CFSv2 databases" (https://data.mendeley.com/datasets/kwcvfr52hp/4, last access: 4 December 2020) (Gramcianinov et al., 2020b)

*Author contributions.* DKS: Conceptualization, Formal analysis, Methodology, Validation, Visualization, Writing — original draft. CBG: Formal analysis, Methodology, Visualization, Writing — original draft. BMC and MD: Writing — review & editing, Supervision.

*Competing interests.* The authors declare that they have no conflict of interest.

*Acknowledgements.* D.K.S was funded during his PhD (CNPq scholarship grants #163120/2015-3 and #201561/2018-2). C.B.G. holds a postdoc scholarship grant #2020/01416–0, São Paulo Research Foundation (FAPESP). The ERA5 products were generated using Copernicus

Climate Change Service Information [2021].The authors are grateful to professor Shuyi Chen's (University of Washington) comments in early stages of this work which improved the manuscript.





## Appendix A: Appendix A

The validation of ERA5 swh results in the wSA with PNBOIA results is summarized in Figure A1. The buoys present measurements between 2012 and 2019 and the swh present high skill (>0.85) (Willmott, 1981), which is also confirmed by high
correlations and a comparable standard ratio between model results and measurements. This validation suggests ERA5 is skillful in simulating the wave variability and consequently the associated surface wind patterns.

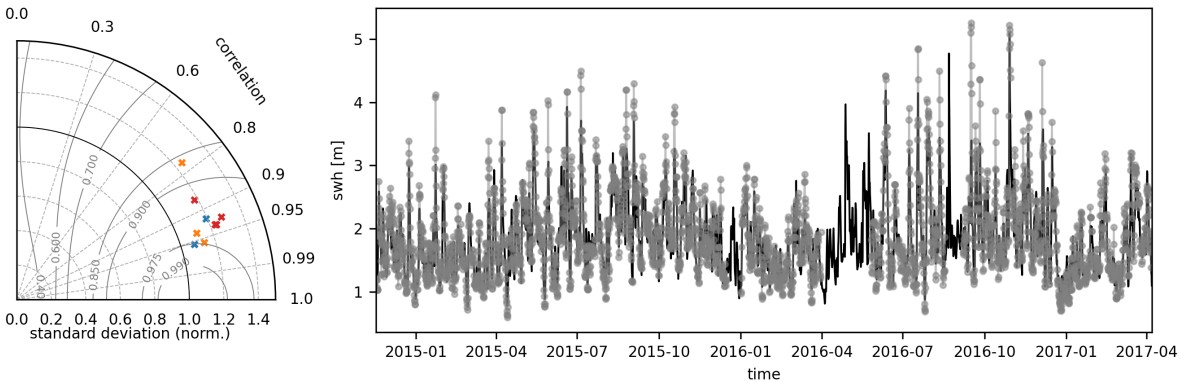

**Figure A1.** The Taylor diagram (left) summarizes the validation of ERA5 swh results with PNBOIA measurements using the results in the interval between 2012-2019. The markers represent different moorings at Santos (red), Santa Catarina (blue), Rio Grande (orange) positions and the gray numbered contours are isolines of skill (Willmott, 1981). Moorings at the same position do not overlap in time. The swh time series (right) exemplifies the ERA5 results (black line) and the measurements (gray dotted line) at Santos position.



## Appendix B: Appendix B

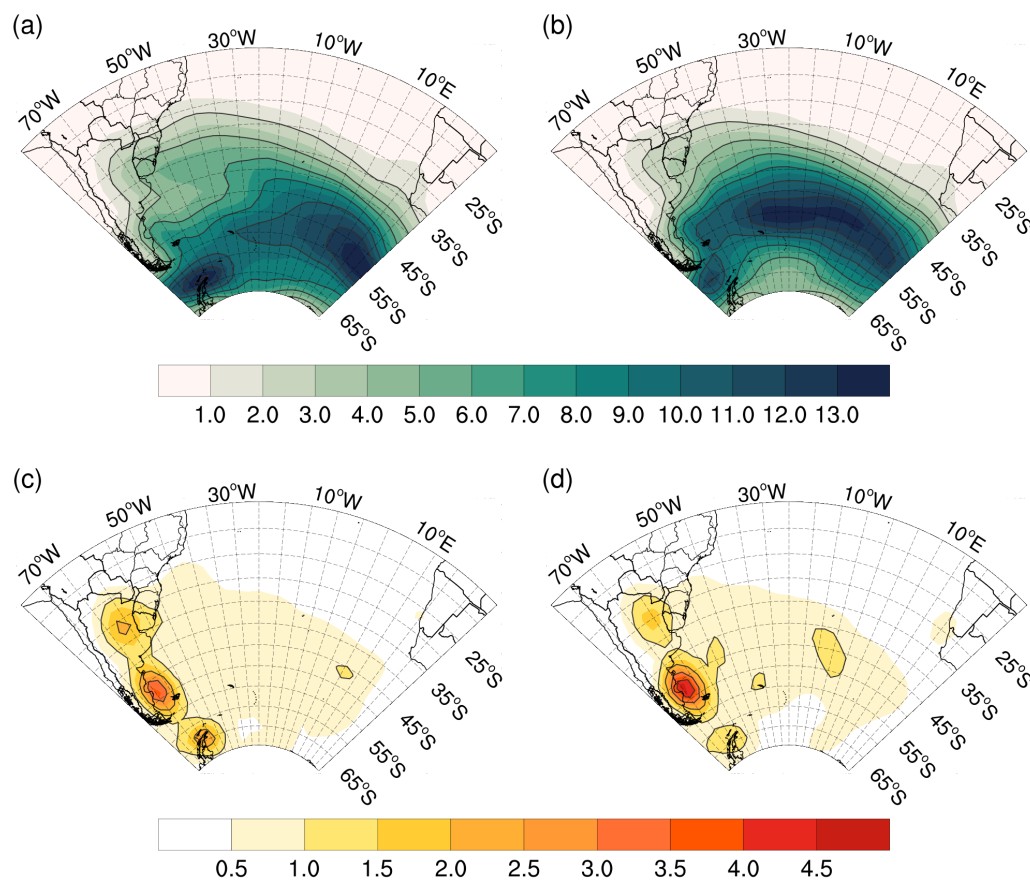

**Figure B1.** Cyclones (a,b) track and (c,d) genesis density in phase (a,c) A and (b,d) B of EOF1 of u10. Dashed contours show 2 density unit intervals for the track and 1 for the genesis. Density unit is track/genesis per month per area, where area is in $10^6$ km$^2$ ($\sim 5°$ spherical cap)

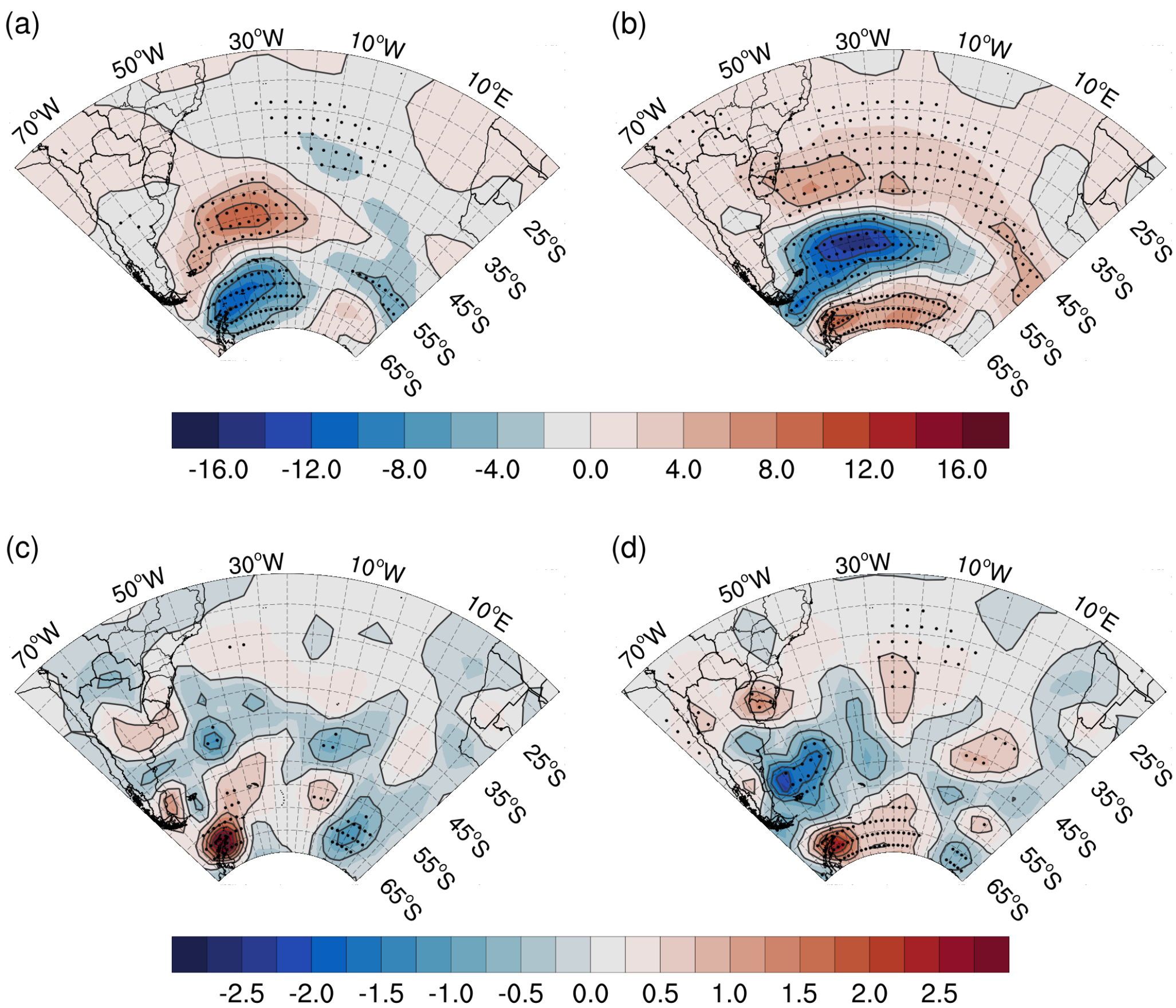

**Figure B2.** Cyclones (a,b) track and (c,d) genesis densities differences between phases A and B of (a,c) EOF1 and (b,d) EOF2 of swh. Differences are calculated using phases A minus B and are also contoured by 5 density unit intervals for the track and 0.5 for the genesis. The black dots indicate where the difference is significant within a 95% confidence level. Density unit is system per year per area, where area is in $10^6$ km$^2$ ($\sim 5°$ spherical cap)

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
