# Peer review of "Intraseasonal variability of ocean surface wind-waves in the western South Atlantic: the role of cyclones and the Pacific South-American pattern"

_Weather and Climate Dynamics, 2021_

## Author Response (AR1)

I would like to submit the revised version of the manuscript "Intraseasonal variability of wind waves in the western South Atlantic: the role of cyclones and the Pacific South-American pattern" authored by Dalton Kei Sasaki, Carolina B. Gramcianinov, Belmiro Castro, and Marcelo Dottori. We accepted the suggestions and corrected the major issues highlighted by the reviewer 2 regarding the inversion of signal in our results. More information and discussion were added to the text to improve the text comprehension and make it more clear. We hope the manuscript can be reconsidered for publication in Weather and Climate Dynamics.

**Reviewer 1**

**The authors examine the relationship between extratropical cyclones, South Pacific - Atlantic intraseasonal variability and extreme significant wave height (swh) values in the western South Atlantic (wSA). Specifically, the authors analyze storm track modulation due to westerlies winds and in particular the intraseasonal component of the Pacific South–American (PSA) mode. Empirical orthogonal function (EOF) analysis of the 10m zonal wind and swh were made using ERA5 reanalysis data in order to assess the westerlies and waves regime in the wSA.**

**The authors found that (1) the intraseasonal signal over the wSA does indeed have a strong role in determining the magnitude of the wave field and that the internal variability of the westerly jets provide the requisite local forcing. The subsequent finding that this intraseasonal variability is in fact linked to the PSA modes is well supported.**

**I found this paper generally well written with a solid analysis and a plausible framework.**

**I would recommend that the authors revise the manuscript to remove the many minor typographical errors and to improve the grammatical errors.**

Reply: We appreciate your revision and comment. We reviewed the new version of the manuscript to correct typos and grammar.

**Reviewer 2**

We sincerely appreciate your comments and careful revision of our work. We tried to address all the highlighted issues and to make the manuscript clearer and more consistent. We believe after this revision the article improved immensely.

It is important to clarify that most of the major issues addressed in this revision were due to a systematic error during the manuscript preparation and we deeply apologize for that. At early stages of this study, we were using an opposite signal orientation in the PC time series and EOF spatial patterns, but during the work evolution we changed the EOF and PC signal to have a more intuitive discussion (notice that the reversal of signal in combined EOF and PC time series does not affect the reconstruction of the signal). Unfortunately, one table and one figure panel were not updated correctly, resulting in some inconsistencies between them and text. These failures do not affect the discussion and conclusion of the work, since the

analysis was made with the correct values/figures. The problem occurred only in the manuscript compilation. After this revision, we double check all information, tables, figures and everything is correct for the reviewed version. All revised lines in this reply refer to the new version, unless specified otherwise.

**Major comments:**

**l. 33-38: you seem to talk about SAM and NAO as though they are primarily interannual indices, but they are really oscillations that we consider relevant on weekly/submonthly timescale. They do of course exhibit interannual, decadal, multidecadal variability. I just think that the wording is perhaps confusing, or maybe I am not understating the point here. ENSO of course is interannual and it can impact SAM/NAO on those timescales, but SAM/NAO are primarily subseasonal. I think this should be corrected throughout the manuscript. It is also confusing when you say that there is no link to SAM here, then discuss later in the results that there is a link?**

Reply: We are sorry for the confusing bits regarding SAM and NAO variability. The reason we put it in terms of interannual variability is the following: our references (Reguero 2015: SAM interannual. Dodet et al. 2013: NAO interannual) refer to the interannual component effect of these indices on the surface gravity waves parameters. We introduced the interannual variability to present a general idea of which scales may be relevant for the regional wave variability. We rewrote the text to make the idea clear (lines 32-38):

"Apart from the seasonal scales, a possible source of predictability of the wave climate could be related to the atmospheric interannual and intraseasonal variability. For instance, the North Atlantic Oscillation interannual variability is relevant in the modulation of significant wave height (swh) in the North Atlantic (Dodet et al., 2010). In the South Atlantic, Pereira and Klumb-Oliveira (2015) observed a significant but weak El Niño Southern Oscillation (ENSO) signal in the swh in wSA. However, so far global studies of wind–wave showed no significant relation between climate indices such as the ENSO or the Southern AnnularMode (SAM) and the wave climate over the wSA, when the interannual component is considered (Godoi et al., 2020; Godoiand Torres Júnior, 2020; Reguero et al., 2015)."

In the discussion, we did not intend to discuss SAM as a cause of variability. In fact, our results show no correlation with SAM and the PCs, which is clarified between lines 256-258 of the reviewed version:

" In the present study, correlation analysis at lag-0 between the PCs (monthly averages) and monthly SAM index (Marshall, 2003) yield values smaller than 0.1, indicating SAM is not relevant regionally. Hence, we concentrate our analysis on the PSA modes."

**164-170; Fig. 5:**

**- phase A in timeseries is when PC is strongly positive (Fig. 5a), hence I would expect the EOF pattern for phase A to be a positive-monopole (i.e. Fig. 5b would then have red colours and 5c blue colours).**

Reply: The swh composite with negative values in Fig 5b is related to the PC in Fig 5a and the EOF pattern in Fig 2c, which also presents negative values. Hence, an increase in the PC value leads to a decrease of the swh fields, as shown in Fig 5b through the composite. We used green and orange in phases A and B of the PC instead of red and blue to avoid an association with positive and negative phases, as the interpretation depends on the EOF spatial pattern. Since this whole idea was not clear, we rewrote the paragraph (line 163-168):

"n the following sections, the intraseasonal relationship between the variability of swh, cyclone genesis and track densities is studied using composites of wave and wind fields based on EOF phases of u10 and swh. We define phase A (B) periods when the PC values are greater (smaller) than 1 standard deviation. These time series have physical meaning only when interpreted in conjunction with the spatial patterns of the EOFs (Fig. 2). For instance, phase A (B) corresponds to positive (negative) values in the time series (Fig. 5a) and the phase combination with the spatial patterns of the EOFs (Fig. 2) generates reconstructed fields with negative (positive) values (not shown), which correspond to the composite fields (Fig. 5b,c)"

**- Also, scale on colourbar seems wrong or maybe scale for PC timeseries is wrong? Is it really in metres?**

Reply:    The unit in the colourbar is correct, as it refers to the field composites (average of the field during a given phase). In order to find the EOFs, we used the covariance matrix and did not scale the principal component to unit variance, hence the relatively large values in the y-axis.  If it was scaled, the dashed lines would coincide with the |1 standard deviation| reference.

**Also, I think it might be better to use u10 in Fig. 5 instead of swh, since you mostly look at composites for u10. Perhaps put current Fig. 5 in Appendix together with all other composites for swh EOFs.**

**- For the sake of consistency, I think you should composite all quantities based on the same variable (i.e. u10, swh, cyclogenesis etc. composited over EOFs of u10; or alternatively over EOFs of swh)**

**- There seem to be differences between composites over EOFs of u10 and EOFs of swh (see below).**

Reply:   We agree that u10 is a better choice in Fig.5 and we replaced the figure. All composite quantities in the text are now consistent with the EOFs of u10. Also, we included the phase composites of both u10 and swh (supplementary material Figs C1,C2), which are helpful in the interpretation of the results. We also added the following text to mention the new figures in line 155: "The phase composites of u10 and swh of the corresponding EOF modes are included in Appendix C".

**Section 3.2: in several places you mention that composites from u10 EOFs are similar/consistent with composites over swh EOFs. I see many differences between the two.**

**- Fig. 6 vs. Fig. B2:**

**- panels (a) largely show opposite sign (where track density is positive in Fig. B2a it is negative in Fig. 6a); and swh composites show weaker anomalies.**

**- panels (b) show a meridional shift between swh and u10 composites (tracks in u10 composite are shifted polewards compared with tracks in swh composite). By how much it is hard to tell. Again, composites over swh show weaker anomalies.**

**- panels (c,d) are somewhat consistent (though it is hard to tell), but anomalies are weaker for swh composites.**

Reply: Panel (a): Thank you very much for the warning. As mentioned before, we had problems during the manuscript compilation and the wrong figure was attached to the panel (Fig. B2a). Panel (b): The meridional shift mentioned in the comment is indeed present within the density track composites. When mentioning the similarities, we refer mainly to the fact that the large-scale signature of EOFs consists of a tripole with a similar spatial structure. In this case, the similar large scale pattern (including the signal) is enough to affirm that they are 'consistent' because the swh field represents the sum of wind-waves (locally forced waves) and the swell component (remotely forced waves). Regarding the 'weaker anomalies' we cited it in line 175 (original document): "The swh related fields are slightly weaker, showing a weaker response of swh EOFs phases, which is expected once this field is also influenced by remote forcing (i.e., swell)". We addressed this behavior to the fact that the wave field is influenced not only by the local wind but also by the remote wind once waves can propagate through the ocean. We clarified it in a new paragraph (line 193-204):

"The density differences based on the EOFs of swh revealed patterns similar to the u10 case (Appendix B, Fig. B2). The stormtrack differences also present a tripole pattern as a consequence of the large-scale wind, similarly to Fig 6. However, these swh related fields are slightly weaker when compared to the u10 case. This weaker response occurs because the swh is integrated by the local (wind-wave) and remote wave (swell) signal (Young, 1999; Chen et al., 2002). Strong winds associated with the cyclones contribute directly to the local generation and development of wind-waves, reflecting in the observed similarities between Figs. 6 and B2. On the other hand, the remote wave signal – the swell – consists of propagating waves generated elsewhere (Alves, 2006; Ardhuin et al., 2009). In other words, the wind and wave fields are partially coupled through wind-waves,which explains the weaker signal in Fig. B2. Also, a meridional shift of a few degrees between the track composites in Fig.2006 and Fig. B2 is present. This shift can be explained by the generation mechanisms of waves within the asymmetric structure of extratropical cyclones. The fully developed sea-state presents higher swh and takes place in the downwind end of the fetch(e.g., Ardhuin and Orfila, 2018), which is usually located northwest from the cyclone center in the wSA (Gramcianinov et al.,2021)."

**- Perhaps the issue is that swh lags behind u10 – e.g. if you do lag-correlations between PCs of u10 and swh you may find a lead-lag relationship between the two. So instead of correlating the two at lag 0 like in Table 1, correlate them for several positive and negative lags, to establish a clearer relationship. If you then lag data**

**accordingly you might then get the "same" results for swh and u10 composites – or just plot general lag-composites. OR the swh and u10 peak in different locations.**

Reply: We believe that the explanation and clarification about this theme were addressed in the last topic, as the comment was based on a figure that we corrected. Notice that waves development after the winds takes only a few hours and this difference is filtered out by the band-pass filter.

**- Another thing I can think of is that EOF1 and EOF2 may not be entirely independent at longer lags (at lag 0 they are by definition uncorrelated) and may represent propagating mode (i.e. if you did a POP analysis [and I am not suggesting you do it] you might find EOF1,2 of u10 to represent the same POP's real and imaginary components). Indeed, PSA (and also SAM) modes are like that and if EOFs1,2 of u10 are related to PSA modes then this can also be a part of the story (i.e. both modes impacting swh at different lags).**

Reply: We appreciate the comment and will take the POP analysis into account in future studies, but we reinforce that the negative correlation was due to the systematic error we corrected.

**- Also, I think that u10/swh A & B composites should be shown over the same regions as cyclone tracks and genesis – that way a link between these quantities can be clearer; i.e. use Fig. 6 type plots also in Fig. 5b,c, & Figs. 7,8.**

Reply: The u10 and swh composites were made to the Southwest South Atlantic, which is the focus of the work. We believe that increasing the domain to evaluate the impacts of the variability in the u10 and swh fields would be detrimental to the regional assessment and compromise our goal. In the case of the cyclone track and genesis, it was necessary to have a larger domain once the cyclone's pattern is more related to large-scale circulation and the wave fields can be influenced by cyclones that occur further south.

**- Note that track densities following wind anomalies are likely consistent with positive baroclinic feedback (such as that presented in Robinson 2000).**

Reply: Very good comment, thank you. We added this information in lines 190-192:

"In both cases, the coupling between track densities and zonal wind anomalies are consistent with positive baroclinic feedback (Robinson, 2000),which shows that the mean flow modifications by baroclinic eddies, i.e., cyclones, reinforce the low-level baroclinicity."

**l. 182: you mention SAM: so do you ultimately find any links to SAM or not?**

Reply: Our results show no correlation with SAM and the PCs, as we answered in an earlier comment.

**l. 201-208: Similar to the above comments: swh and u10 seem to be out of phase – perhaps plotting both of them on the same plot (one in contours and one in shading) could help you (or me) whether they are out of phase and by how much. Again, there is likely a lead-lag relationship or they are simply peaking in different locations.**

Reply: We apologize again for the signal mistake in Fig. B2(a). We hope this question is solved with the correct figure. In any case, the filter we applied (line 125-129) removed propagating signals from swh and u10, which implies the variables are peaking at different locations. This is expected due to the swell component in the swh, which does not depend on the local wind, as mentioned already in some comments above.

**l. 216-217: I can also see SW-NE orientation south-west of SBB, which makes me wonder if this is what brings high shww to SBB?**

Reply: This is an interesting observation, thank you. It is difficult to relate the observed pattern in Fig. 7 and 8 with the high shww in the SBB (Fig. 9) because the shww is the locally forced fraction of the swh, so the swh field south-west of SBB would not influence the shww in the SBB. The above-mentioned SW-NE orientation indicates the fetches orientation in the region - which is explored more further on in the manuscript. We added a comment in lines 218-220:

"However, the SW-NE orientation of the anomalies is more evident in the extreme composites, which indicates the dominant orientation of the wave generation fetches in the wSA (e.g., Campos et al., 2018; Gramcianinovet al., 2021)."

**Fig. 9: I think I can see cyclone-anticyclone (trough-ridge) pairs in all panels, but the exact position, orientation and magnitude differ. For example, Fig. 9a,d have the pair oriented along the S. America coast (i.e. SW-NE), but in Fig. 9b,c the orientation is perpendicular to the coast (i.e. NW-SE).**

**- Perhaps you could think about a future study where you could do a regime perspective (e.g. using K-means) to really classify different regimes that cause this swh. [just a suggestion for future work]**

Reply: The suggestion of using a regime perspective is really great. Actually, we had similar ideas when we first saw these patterns and we are already working on it in a forth-coming study. We added some comments (line 242-248) about the cyclone-anticyclone patterns (Fig. 9), which have been proved to play a big role in extreme wave generation:

"Composites of transient-related events are often noisy since the cyclone's position and associated features (e.g.,cold and warm fronts) are mobile. For this reason, the wind patterns in Fig. 9 do not present a closed cyclonic circulation, but a trough instead. It is also possible to see cyclone-anticyclone (trough-ridge) pairs with different orientations, positions, and magnitudes.This happens due to the rich variety of atmospheric patterns associated with extreme waves in the wSA (da Rocha et al., 2004; Solari and Alonso, 2017; Gramcianinov et al., 2020c). In fact, Gramcianinov et al. (2020c) showed that the presence and relative position of the anticyclone to the cyclone may contribute to the extreme wave event generation by enlarging the fetch and increasing the wind speed"

**Fig. 11 and discussion around it:**

**- The years/dates discussed in text and Fig. caption do not match panel titles. So I am not sure if the panels are wrong, or their titles.**

Reply: We are sorry for the mismatchment, the panels were addressing another period and we corrected it in Fig. 11.

**- I also find it hard to follow what feature the authors are talking about – I suggest circling the features you discuss (or drawing a line along the wave train)**

Reply: We added lines in Fig. 11, as suggested, and also altered the text (lines 291-293) and replaced specific dates to visual markers:

"Green dashed lines in Fig 11 exemplify positive signals in between 180◦W and 90◦W propagating towards 30◦W. These signals take up to four month to cross the South-Pacific domain. Other features can be noted as westward propagating signals (light green dotted lines), "

**Overall, I think that some lead-lag relationships are missing, and that once those are established everything will make sense.**

Reply: We appreciate all the comments. As explained before, the issues regarding the lead-lag relationship were related to a panel that didn't present the right information. We hope that with the changes, corrections, and clarifications, the proposed relations make more sense now.

**Other comments**
**Rephy:** We accepted all minor corrections. Here we reply to the remaining questions.

- **"wind waves" – are you referring to storm surge or something else? Please clarify in the introduction.** Reply: Thank you for the comment. Wind-waves are gravity waves generated by the wind, with a larger frequency than storm surge. We added a brief explanation in the first paragraph.
- **l. 121: by "mean daily climatology"** – have you smoothed it or is it raw mean? [just checking] Reply: The mean daily climatology is simply the climatological mean of each day of the year over the entire ERA5 dataset. In other words we have ~365 daily climatological means.
- **Fig. 2: u10 EOFs look more tilted than EOFs of swh; the location of negative lobes of u10 EOF1,2 are where SAM can have an impact (which is somewhat mentioned later in the text);** Reply: We found no correlation between SAM and the EOFs, as commented in an earlier reply.
- **Table 1: I am little bit confused by the correlations – EOF1 u10 vs EOF1 swh is a positive correlation; but other u10 and swh correlations are negative, suggesting anti-correlation (i.e. positive u10 mode related to negative swh mode – strong for EOF2). Table caption – if correlations are computed at "lag-0" please specify it.** Reply: All correlations are computed in lag-0. Actually both the signals in the column of EOF2 u10 were inverted, as explained in the replies above.

- **Fig. 4: Is there no red-noise-like low-frequency "peak" because you consider periods shorter than 16 years or? I would expect red-noise like behaviour at low frequencies.** Reply: There is little to no red-noise in the signal (the time series of EOF). This was surprising for us as well at first, but it makes sense when we consider the results from Reguero et al. 2015. These authors made several global analyses and showed no significant signal in interannual scales with respect to several climate indexes in the South Atlantic. The South America continent probably blocks/filters incoming surface wave signals (and u10) from the Pacific, which carry interannual information. Also, when we evaluate the time series anomalies at several spatial positions of u10, v10 and swh (not shown) there is no 'structure' that resembles periods higher than 1-2 years (frequencies lower than 1-0.5year$^{-1}$). This was supported by the time series in Fig. 3, where the results almost behave as white noise, which is coherent with the wavelets figures.
- **l. 220-229: you mention cyclones in different locations, but I also see anticyclone-like features over the continents in some cases and over the sea in other cases.** Reply: We hope we replied to this comment in the questions above and with the addition of the text in the lines 242-248 of the revised manuscript.
- **l. 223-224: you see a cyclone to the southwest of SBB in Fig. 9c? Is it outside the map's bounds (i.e. not shown)?** Reply: The cyclone center positions on Fig. 9 are marked with black dots, but due to the positional spread the wind composites don't show the cyclone clearly but a trough instead. The trough plus the cyclone center locations supported the discussion between lines 226 and 233, and we added a comment on that in lines 242-248
- **l. 292-3: As mentioned under major comments: swh and u10 modes can be out of phase.** Reply: We hope this was solved in the major comments replies.
- **l. 344-348: I know the authors find no tropical links, but the impact from PSA on genesis reminds me of the paper by Schemm et al. 2018, who showed that N. Atlantic genesis location depended on ENSO phase (here it may depend on PSA phase).** Reply: Thanks for the reference, this is a very interesting paper. It probably will be helpful in future studies in explaining physical processes that influence the propagation of the PSA signal from the Pacific to the Atlantic and ultimately the cyclogenesis and wave fields.

---

## Referee Report (RR1)

Review for "Intraseasonal variability of wind waves in the western South Atlantic: the role of cyclones and the Pacific South-American pattern" by Sasaki et al.

I acknowledge most of the authors responses – I believe I did not read carefully in places, but I am glad we identified so errors too, so the manuscript is largely ready to go. I still have a few questions/comments that I would like to see addressed before publication.

Comments:

l. 73: atmosphere -> atmospheric

Table 1: I am still confused by the negative values in the table. Are they a consequence of different signs of the EOFs and hence of the PCs (they do have arbitrariness in sign)? E.g. EOF1 swh & EOF1 v10 correlation is negative. Is this because EOF1 swh is e.g. a positive monopole and EOF1 v10 is a negative monopole? Therefore correlation of the corresponding PCs is negative? But it means that stronger swh is related to stronger v10 as alluded to in the text?

- If this is true then I find this confusing. I would usually choose a sign of the EOF (e.g. positive monopole in an EOF) and then multiply PCs and EOFs by (-1) if the sign in the EOF is opposite. That way I can avoid this confusion. I recommend doing this, since I think this would make it much easier for the reader.
    - Given Fig. 5 I guess I would choose negative monopole to keep the additional work to a minimum.
    - Also, if this issue only applies to the correlations in Table 1 and you know all correlations would be positive if you defined all EOFs in a consistent way then you can just drop minuses in the Table 1 and that's that.
- If this is not true, I would recommend addressing this in the text – i.e. saying negative correlations mean stronger swh, weaker v10; positive correlations mean stronger swh & stronger v10 (v10 & swh here are just examples; feel free to adjust).

l. 199: "the wind and wave fields are partially coupled through wind waves" – do you mean they are "only partially" coupled? Since the amplitude is small and remote effects lower the links?

l. 288: n -> in

l. 291: Green dashed lines -> Thin green dashed lines

l. 293: light green dotted lines -> thick green dotted lines

l. 285-294: You mention westward propagating waves – are periods that show westward propagation related to e.g. larger (more planetary) waves, rather than synoptic waves (in scale)? Or is it largely same waves propagating eastward/westward? If the latter then no need to add any sentences.

l. 297-303: You say that phase A has stronger wind, but weaker storm track, lower swh? But I thought you established a positive baroclinic feedback where stronger winds also have stronger storm track. Am I missing something again? Also because you then continue on saying "on the other hand", phase B has weaker winds, weaker storm track ………..

Fig. 12: This figure is not mentioned anywhere thus redundant – please remove it or discuss it . I also find the figure confusing – there are linear relationships, but one shows negative the other positive regression coefficients. Is that again due to PCs having inconsistent signs?

---

## Author Response (AR2)

Dear Dr. Hassanzadeh

I would like to submit the revised version of the manuscript "Intraseasonal variability of wind waves in the western South Atlantic: the role of cyclones and the Pacific South-American pattern" authored by Dalton Kei Sasaki, Carolina Barnez Gramcianinov, Belmiro Castro, and Marcelo Dottori. We accepted the suggestions and corrected the issues highlighted by you and reviewer 2 and hope the improved and revised manuscript can be reconsidered for publication in Weather and Climate Dynamics. Your (and the reviewer's) comments are marked in bold characters, while our answers are marked by plain characters. We deeply appreciate all comments and suggestions.

Sincerely,

Dalton Kei Sasaki

**Reviewer 1**

Dear Authors,
Thank you for addressing the earlier comments and suggestions provided by both reviewers. Reviewer 2, while overall happy with the revised version, still has a few important comments that should be fully addressed. In particular, comment 2 (about Table 1) and the last two comments need your careful consideration.

Furthermore, while the paper is overall well-written and easy to follow, I believe that the text still needs some polishing. This is mostly to improve clarity and fix typographical errors and inconsistencies, and to ensure that all acronyms are defined the first time they are used. In non-public comments, there are some examples based on my own reading of the paper, but keep in mind that this is by no means an exhaustive list. Please carefully go through the paper to address similar and other typographical issues.

Thank you for submitting your interesting work to WCD, and I am looking forward to receiving the revised manuscript.

Sincerely yours,

Pedram Hassanzadeh

**Non-public comments to the Author:**
Dear authors,
Here are some suggestions based on my own reading of the manuscript. Some are just suggestions (and feel free to ignore) but some are typographical errors. Please carefully go through the paper and fix similar issues.

**Title: it may not be clear to the reader what "wind waves" mean. I suggest using something like "ocean surface wind waves" in the title.**
We accepted the suggestion, thank you.

**Abstract: same as in the title, please make sure it is clear that this is about ocean surface winds; e.g., use "extreme significance wave heights (swh) at the ocean surface in the …."**
We accepted the suggestion, thank you.

**Line 3 and other places (e.g., lines 24, 31): "wind-wave climate" or "wave climate" – it might be better to say "climatology of wave wind" and "climatology of waves"**
In the text, sometimes wave climate is related to extreme cases. In these excerpts, we substituted 'climate' for 'events'. In some cases we did really mean climate, but we understand that climatology is better suited in some contexts and we changed it accordingly. Thank you for the suggestion.

**Wind-wave vs wind wave: is not consistent between tile, lines 3, lines 16 etc**
We corrected it, thank you.

**Lines 3 and 4: time scales vs time-scales: use one consistently throughout the paper**
We corrected it, thank you.

**Line 11: u10 should be defined, for example in line 7 after 10m zonal wind**
We corrected it, thank you.

**Line 16: social-economic --> socio-economic**
We corrected it, thank you.

**Line 36: "global studies of wind–wave showed" --> "global studies of waves showed" given the statement in line 16**
We corrected it, thank you.

**Line 52: is PC already defined somewhere? It is later defined in page 5.**
We did define it after line 52. We corrected the text, thanks.

**Line 97: maybe I missed it, but what is the meaning of ^100 here? Is it explained somewhere?** It refers to the global centennial  part. Since it is not used throughout the text it does look unnecessary and we removed it.

**Line 125: EOF is first use in line 118 and should be defined there.**
We corrected it, thank you!

**Line 367: (2) Which are … --> (2) What are …**
We corrected it, thank you!

**Line 375: naval structure --> naval structures**
We corrected it, thank you!

**- in several places, enumerated items are presented in line. To improve clarify, e.g., on page 3 (line 70 and onward), i suggest to make a list with each item appearing on a separate line.**
We changed the list of the items on separate lines, except for the enumeration case in the first paragraph of the conclusion. We chose to not modify the conclusion part since it won't improve the understanding and it doesn't look very good.

**Review for "Intraseasonal variability of wind waves in the western South Atlantic: the role of cyclones and the Pacific South-American pattern" by Sasaki et al.**

**I acknowledge most of the authors responses – I believe I did not read carefully in places, but I am glad we identified so errors too, so the manuscript is largely ready to go. I still have a few questions/comments that I would like to see addressed before publication.**

**Comments:**
**l. 73: atmosphere -> atmospheric**

**Table 1: I am still confused by the negative values in the table. Are they a consequence of different signs of the EOFs and hence of the PCs (they do have arbitrariness in sign)? E.g. EOF1 swh & EOF1 v10 correlation is negative. Is this because EOF1 swh is e.g. a positive monopole and EOF1 v10 is a negative monopole? Therefore correlation of the corresponding PCs is negative? But it means that stronger swh is related to stronger v10 as alluded to in the text?**

- **If this is true then I find this confusing. I would usually choose a sign of the EOF (e.g. positive monopole in an EOF) and then multiply PCs and EOFs by (-1) if the sign in the EOF is opposite. That way I can avoid this confusion. I recommend doing this, since I think this would make it much easier for the reader.**
    - **Given Fig. 5 I guess I would choose negative monopole to keep the additional work to a minimum.**
    - **Also, if this issue only applies to the correlations in Table 1 and you know all correlations would be positive if you defined all EOFs in a consistent way then you can just drop minuses in the Table 1 and that's that.**
- **If this is not true, I would recommend addressing this in the text – i.e. saying negative**
  **correlations mean stronger swh, weaker v10; positive correlations mean stronger swh & stronger v10 (v10 & swh here are just examples; feel free to adjust).**

Indeed the EOF1 swh and EOF1 v10 are monopoles with opposite values. *We are changing the table's values to make it coherent with the text, we appreciate the feedback.*

**l. 199: "the wind and wave fields are partially coupled through wind waves" – do you mean they are "only partially" coupled? Since the amplitude is small and remote effects lower the links?**

Before answering the question, we would like to make a comment. We realized that we used wind-wave with two different meanings throughout the text. In the first case, it means 'waves forced by winds' which is used in the title, abstract and introduction. In the second case, wind-waves are actually the locally forced waves (which are also known as wind-sea).

After clarifying this point, we point that waves forced by winds present, in theory, two components - the swell and the wind-sea. Between these components, only wind-sea are coupled to the winds. The swell is a result of the non-linear interaction of waves with different frequencies and directions, which produces a wave field component independent from the wind. Obviously, these are conceptual models that help understand these processes. Hence, the wind and wave fields may be partially coupled through wind-waves.

We are sorry for the confusion regarding the wind-wave meaning. We changed the text and updated the words to wind-sea and wind-wave.

**l. 288: n -> in**
We corrected it, thank you!

**l. 291: Green dashed lines -> Thin green dashed lines**

We accepted the suggestion, thank you!

**l. 293: light green dotted lines -> thick green dotted lines**
We accepted the suggestion, thank you!

**l. 285-294: You mention westward propagating waves – are periods that show westward propagation related to e.g. larger (more planetary) waves, rather than synoptic waves (in scale)? Or is it largely same waves propagating eastward/westward? If the latter then no need to add any sentences.**
By the Hovmöllers (Fig. 11), these westward propagating waves seem to have the same phase speed as eastward waves, so we understand that they are mainly synoptic waves in scale. Of course, a more robust analysis needs to be done in future research.

**l. 297-303: You say that phase A has stronger wind, but weaker storm track, lower swh? But I thought you established a positive baroclinic feedback where stronger winds also have stronger storm track. Am I missing something again? Also because you then continue on saying "on the other hand", phase B has weaker winds, weaker storm track ...........**
The reviewer's comment is correct. We described phase B incorrectly after the expression 'on the other hand'. We rewrote the text with the right description of it (lines 304-307):

*On the other hand, the intensification of the surface winds, observed in phase B, may play a role in the strengthening of baroclinic wave growth, as long as it reflects directly to the upper-level jet and baroclinicity \citep{Hoskins1993,Ambrizzi1997}. In this situation, more frequent and more intense cyclones are expected to develop in the region representing, thus, a more efficient wave generation mechanisms.*

**Fig. 12: This figure is not mentioned anywhere thus redundant – please remove it or discuss it . I also find the figure confusing – there are linear relationships, but one shows negative the other positive regression coefficients. Is that again due to PCs having inconsistent signs?** - ok
The PCs with weird signs are to blame for the confusing PCs signals. In summary, low pressure systems (Figure a - associated with throughs) strengthens the zonal wind in the EOF u10 on the monopole region. In Figure b, the center of the northern node of the dipole is enhanced by stronger u10 winds. This figure was meant to further emphasize the physical consistency of the EOFs and actual physical variables and is redundant with other results and for some reason the text that explained it was missing. We are removing the figure, as we believe it doesn't add much to the results or the discussion

---

## Author Response (AR3)

Dear Dr Hassanzadeh,

We would like to submit the revised version of the manuscript "Intraseasonal variability of ocean surface wind-waves in the western South Atlantic: the role of cyclones and the Pacific South-American pattern" authored by Dalton Kei Sasaki, Carolina Barnez Gramcianinov, Belmiro Castro, and Marcelo Dottori.

The final reviewer's comments pointed to the signal in Table 1 of the correlation between EOF2 v10 and EOF2 swh (0.12). We switched it back to -0.12, which is the correct value. We deeply appreciate all the comments and suggestions, for this reason we added the following sentence to the acknowledgements: `The authors are also thankful to the anonymous reviewers for their careful reading and relevant suggestions.`

Sincerely,

Dalton Kei Sasaki